# NPAS4 recruits CCK basket cell synapses and enhances cannabinoid-sensitive inhibition in the mouse hippocampus

Andrea L Hartzell[1,2], Kelly M Martyniuk[2†], G Stefano Brigidi[2], Daniel A Heinz[2,3], Nathalie A Djaja[2], Anja Payne[1,2], Brenda L Bloodgood[1,2]*

[1]Neuroscience Graduate Program, Center for Neural Circuits and Behavior, University of California San Diego, San Diego, United States; [2]Division of Biological Sciences, Section of Neurobiology, Center for Neural Circuits and Behavior, University of California San Diego, San Diego, United States; [3]Biological Sciences Graduate Program, Center for Neural Circuits and Behavior, University of California San Diego, San Diego, United States

**Abstract** Experience-dependent expression of immediate-early gene transcription factors (IEG-TFs) can transiently change the transcriptome of active neurons and initiate persistent changes in cellular function. However, the impact of IEG-TFs on circuit connectivity and function is poorly understood. We investigate the specificity with which the IEG-TF NPAS4 governs experience-dependent changes in inhibitory synaptic input onto CA1 pyramidal neurons (PNs). We show that novel sensory experience selectively enhances somatic inhibition mediated by cholecystokinin-expressing basket cells (CCKBCs) in an NPAS4-dependent manner. NPAS4 specifically increases the number of synapses made onto PNs by individual CCKBCs without altering synaptic properties. Additionally, we find that sensory experience-driven NPAS4 expression enhances depolarization-induced suppression of inhibition (DSI), a short-term form of cannabinoid-mediated plasticity expressed at CCKBC synapses. Our results indicate that CCKBC inputs are a major target of the NPAS4-dependent transcriptional program in PNs and that NPAS4 is an important regulator of plasticity mediated by endogenous cannabinoids.
DOI: https://doi.org/10.7554/eLife.35927.001

*For correspondence:
blbloodgood@ucsd.edu

Present address: †Department of Neuroscience, Columbia University, New York, United States

Competing interests: The authors declare that no competing interests exist.

## Introduction

Immediate-early gene transcription factors (IEG-TFs) are expressed in response to sensory experiences and are routinely used to identify task-relevant neurons (*Bullitt, 1990*; *Guenthner et al., 2013*; *Renier et al., 2016*; *Ye et al., 2016*), including those associated with memory formation and behavioral plasticity (*Alén et al., 2013*; *Cai et al., 2016*; *Cowansage et al., 2014*; *Garner et al., 2012*; *Mayford and Reijmers, 2015*). In spite of their wide-spread use as tools, surprisingly little is known about how IEG-TFs alter connectivity between specific neuron subtypes, influence plasticity, or impact circuit function (*Minatohara et al., 2015*).

The IEG-TF neuronal PAS domain protein 4 (NPAS4) is expressed exclusively in response to membrane depolarization (*Lin et al., 2008*), explicitly linking this expression to the activity history of the neuron. Behaviorally induced NPAS4 directs the reorganization of inhibition along the somato-dendritic axis of hippocampal pyramidal neurons (PNs), enhancing somatic inhibition and reducing inhibition in the proximal dendrites (*Bloodgood et al., 2013*). NPAS4 is therefore poised to convert transient increases in neuronal activity into long-lasting changes in how PNs are functionally embedded in the local inhibitory circuit. However, we do not know whether the expression of NPAS4 in PNs leads to the regulation of synapses made by specific inhibitory neuron subtypes.

Inhibitory neuron subtypes are highly heterogeneous, each with distinct functions within their local circuit. In CA1 of the hippocampus, somatic inhibition is provided by cholecystokinin (CCK)- and parvalbumin (PV)-expressing basket cells (BCs), which have unique and complementary roles in gating PN output. This provides a straightforward system in which to ask whether NPAS4 regulates select subtypes of inhibitory inputs. Fast-spiking PVBCs provide reliable, precisely timed neurotransmission in response to predominantly feedforward synaptic input (*Glickfeld and Scanziani, 2006*), positioning them to finely regulate the timing of PN action potential (AP) firing (*Pouille and Scanziani, 2001*). By contrast, CCKBCs elicit slower and less reliable, asynchronous inhibition (*Daw et al., 2009*; *Hefft and Jonas, 2005*) in response to the summation of both feedforward and feedback synaptic input (*Glickfeld and Scanziani, 2006*). Importantly, CCKBCs express a variety of neuromodulatory receptors, including cannabinoid receptors (CB1Rs) that are localized to their axon terminals (*Dudok et al., 2015*; *Glickfeld et al., 2008*; *Katona et al., 1999*). Indeed, activation of CCKBC CB1Rs by endocannabinoids, which are released from postsynaptic PNs, underlies depolarization-induced suppression of inhibition (DSI) (*Wilson and Nicoll, 2001*), a form of retrograde signaling that confers onto active PNs a transient window of increased excitability and plasticity (*Carlson et al., 2002*; *Chevaleyre and Castillo, 2004*; *Zhu and Lovinger, 2007*). Activity-driven NPAS4 expression increases somatic inhibition onto PNs, but it is unknown whether this synaptic regulation is interneuron subtype-specific. Given the marked differences between PV- and CCKBCs, the functional significance of experience-driven modulation of somatic inhibition will be determined by the subtype(s) of basket cell synapses that are regulated by NPAS4 expression.

Using behavioral manipulations in combination with electrophysiological, pharmacological, and anatomical approaches, we show that novel sensory experiences selectively increase the number of inhibitory synapses made by individual CCKBCs onto CA1 PNs through an NPAS4-dependent mechanism. Moreover, we find that this interneuron subtype-specific circuit change strongly enhances DSI expression by active PNs.

## Results

### NPAS4 is expressed in CA1 PNs in response to environmental enrichment

We manipulated the sensory experiences of juvenile wildtype (WT) mice by housing littermates in an enriched environment (EE), which consisted of a running wheel and several novel objects that were regularly refreshed (*Figure 1A*, see 'Materials and methods' for details). After four days in EE, hippocampi were removed, sectioned, and immunostained with antibodies recognizing NPAS4 and the neuronal marker NeuN. Comparable immunostaining was performed on sections from age-matched mice housed in unenriched, standard environments (SE). We observed a significant increase in NPAS4-positive neurons in CA1 from mice allowed to explore an EE relative to those maintained in an SE (*Figure 1B and C*; SE: $3.4 \pm 0.6\%$, EE: $11.2 \pm 0.5\%$; U=0, p<0.001, Mann-Whitney U Test), similar in magnitude to what has been reported previously (*Bloodgood et al., 2013*) and indicating that many CA1 cells had been active recently. As NPAS4 protein is rapidly produced and degraded (*Lin et al., 2008*), this result probably significantly underestimates the percentage of neurons that expressed NPAS4 over the duration of the four days in EE.

Recent studies indicate functional distinctions between PNs in the superficial (closest to stratum radiatum, SR) and deep (closest to stratum oriens, SO) sublayers of CA1 (*Geiller et al., 2017*), including gene expression profiles (*Cembrowski et al., 2016*), firing patterns (*Baimbridge et al., 1991*), and connectivity with excitatory (*Masurkar et al., 2017*) and inhibitory neurons (*Lee et al., 2014*; *Valero et al., 2015*). We therefore asked whether superficial and deep CA1 PNs express NPAS4 equivalently in response to exploration of an EE. We observed no significant effect of sublayer on NPAS4 expression as well as no significant interaction between CA1 sublayer and housing environment (*Figure 1—figure supplement 1*; Superficial EE: $11.3 \pm 0.8\%$ of neurons, Deep EE: $10.3 \pm 1.0\%$ of neurons, Sublayer: $F_{(1,32)}=1.86$, p=0.18, Interaction: $F_{(1,32)}=0.04$, p=0.84, two-way ANOVA), suggesting that experience-driven NPAS4 expression is similar among superficial and deep CA1 PNs.

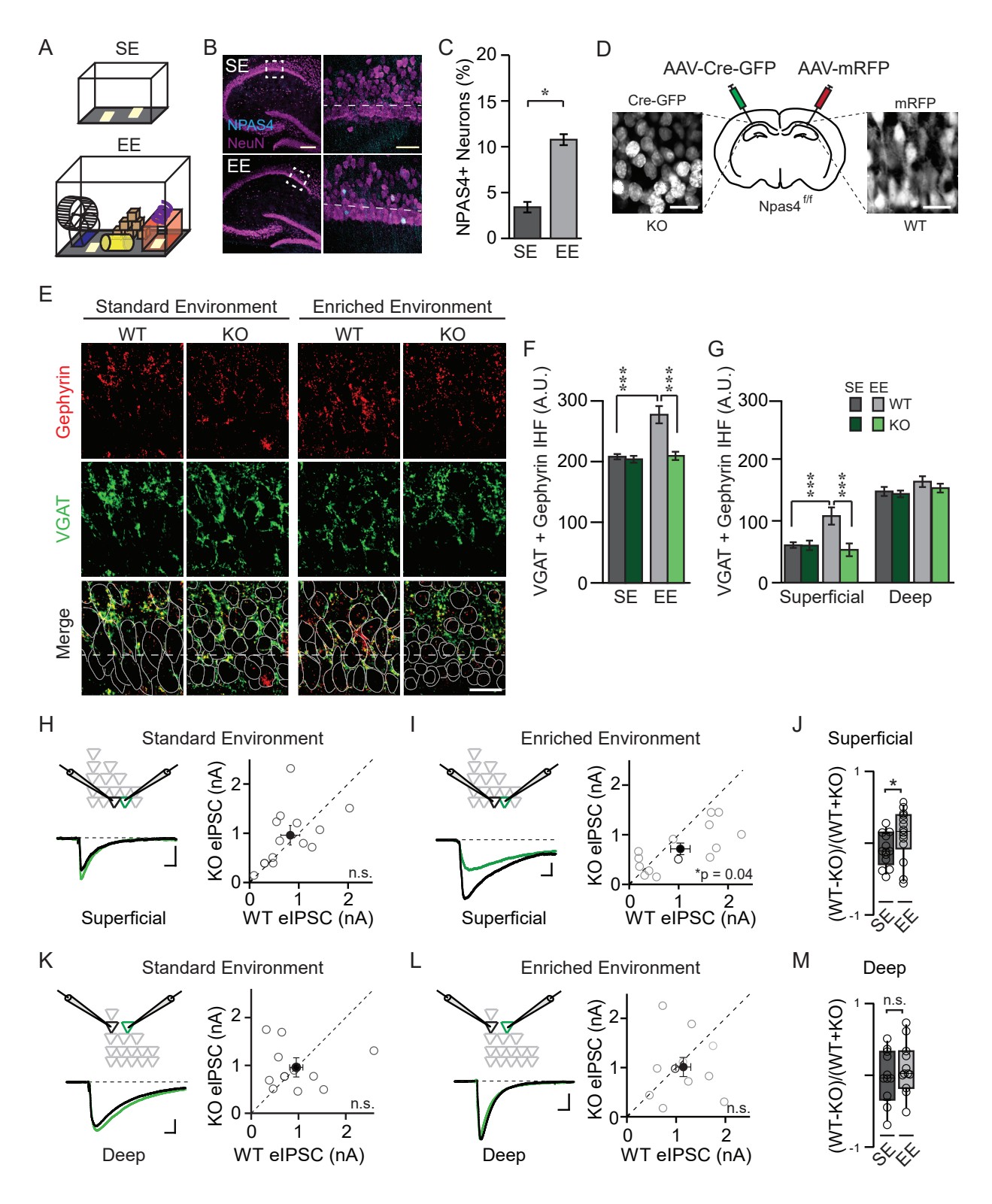

**Figure 1.** NPAS4 enhances somatic inhibition onto superficial CA1 pyramidal neurons. (A) Cartoon of standard environment (SE) and enriched environment (EE). (B) Confocal images of mouse hippocampi from SE (*top*) and EE (*bottom*) stained with antibodies recognizing NPAS4 (*cyan*) and NeuN (*magenta*). Left: box indicates the region imaged at high magnification and quantified for NPAS4-positive neurons. Scale bar = 200 µm. *Right:* higher magnification of the boxed regions shown in the images on the left. Dashed lines indicate the boundary of superficial CA1 (25 µm from edge of

*Figure 1 continued on next page*

*Figure 1 continued*

pyramidal cell layer). Scale bar = 50 µm. (**C**) Quantification of NPAS4-positive neurons in CA1 in mice from SE and EE (n = 10 sections over 3 mice per condition). * indicates p<0.05. (**D**) Schematic of stereotaxic adeno-associated virus (AAV) infection of CA1 in *Npas4$^{f/f}$* mice. The hemispheres of each mouse were randomly assigned to control or knock-out (KO) conditions and hippocampi were infected accordingly with AAV-Cre-GFP (*left*) and AAV-RFP (*right*) in the contralateral hemisphere. (**E**) Representative images stained with antibodies recognising gephyrin (*red*) or presynaptic vesicular GABA transporter (VGAT) (*green*) and merged immunohistochemistry (IHC) from WT and *Npas4* KO hemispheres from mice housed in SE and EE. Gray dotted lines in merged images represent cell body outlines in the WT condition (mRFP expression) or in cell nuclei in the *Npas4* KO condition (Cre-GFP expression). Scale bar = 20 µm. (**F**) Quantification of inhibitory synapses (overlap of VGAT and gephyrin immunofluorescence) in WT and *Npas4* KO hemispheres from mice housed in SE and EE (SE: n = 10–11 sections from a total of 5 mice; EE: n = 9–11 sections from a total of 6 mice). *** indicates p<0.001. (**G**) Quantification of inhibitory synapses (overlap of gephyrin and VGAT immunofluorescence in superficial and deep sublayers of CA1 in WT and *Npas4* KO hemispheres from mice housed in SE and EE (SE: n = 7 sections from a total of 3 mice, EE: n = 4–6 sections from a total of 3 mice). *** indicates p<0.001. (**H**) Standard environment: example eIPSC from WT (*black*) and KO (*green*) PNs in superficial CA1 (*left*). Pairwise comparison of eIPSCs recorded in neighboring WT and *Npas4* KO neurons (*gray, right*, n = 11 pairs). (**I**) Enriched environment: example eIPSC from WT (*black*) and KO (*green*) PNs in superficial CA1 (*left*). Pairwise comparison of eIPSCs recorded in neighboring WT and *Npas4* KO neurons from mice housed in EE (*gray, right*, n = 14 pairs). (**J**) Normalized eIPSCs recorded from WT and KO pairs across SE and EE for superficial CA1. * indicates p<0.05. (**K**) Standard environment: example eIPSC from WT (*black*) and KO (*green*) PNs in deep CA1 (*left*). Pairwise comparison of eIPSCs recorded from neighboring deep WT and *Npas4* KO PNs (*right*, n = 11 pairs). (**L**) Enriched environment: example eIPSC from WT (*black*) and KO (*green*) PNs in deep CA1 (*left*). Pairwise comparison of eIPSCs recorded from neighboring deep WT and *Npas4* KO PNs (*right*, n = 11 pairs). In (H–I) and (K–L), open circles indicate individual pairs, the darker circle is the example trace, and the filled circle indicates the mean ± SEM. (**M**) Normalized eIPSCs recorded from WT and KO pairs across SE and EE for deep CA1.

DOI: https://doi.org/10.7554/eLife.35927.002

The following source data and figure supplements are available for figure 1:

**Source data 1.** NPAS4 enhances somatic inhibition onto superficial CA1 PNs.
DOI: https://doi.org/10.7554/eLife.35927.005
**Figure supplement 1.** Superficial and deep CA1 PNs express NPAS4 equivalently after exploration of an EE.
DOI: https://doi.org/10.7554/eLife.35927.003
**Figure supplement 1—source data 1.** NPAS4 expression in superficial and deep CA1.
DOI: https://doi.org/10.7554/eLife.35927.004

## NPAS4 underlies an experience-dependent enhancement of somatic inhibition in superficial CA1

We visualized somatic inhibitory synapses in CA1 using immunohistochemistry and made the unexpected observation that NPAS4 expression most prominently affects inhibitory synapses onto superficial CA1 PNs. The CA1 region of the hippocampus from *Npas4$^{f/f}$* mice was densely infected with adeno-associated viruses (AAV) encoding mRFP (AAV-mRFP) in one hemisphere and Cre-GFP (AAV-Cre-GFP) in the other (assigned randomly), enabling within-animal comparisons of wildtype (WT) and *Npas4* knockout (KO) hemispheres (*Figure 1D*), respectively. Four days after surgery, allowing time for virus expression, *Npas4* excision, and degradation of preexisting NPAS4 protein, mice were housed for an additional four days in SE or EE, then their hippocampi were removed, fixed, and sectioned. To detect inhibitory synapses, sections in which > 95% of PNs were infected were stained with antibodies recognizing presynaptic vesicular GABA transporter (VGAT) protein and the inhibitory postsynaptic scaffolding protein gephyrin. The overlap of immunofluorescence within the pyramidal cell layer was quantified as a proxy for somatic inhibitory synapses.

We compared immunofluorescence between WT and KO hemispheres in mice housed in SE and EE using a two-way ANOVA and observed a significant effect of genotype and housing as well as a significant interaction effect (*Figure 1F*; Genotype: $F_{(1,37)}$=17.96, p=0.0001; Housing: $F_{(1,37)}$=16.38, p=0.0003; Interaction: $F_{(1,37)}$=11.46, p=0.0017, two-way ANOVA). In mice housed in SE, WT and KO hemispheres had equivalent immunofluorescence (*Figure 1E and F*; WT: 208.0 ± 4.4 arbitrary units (AU), KO: 203.8 ± 5.6 AU; p>0.05, two-way ANOVA with Bonferroni post hoc test), indicating that the somatic inhibition of WT and KO neurons is similar in mice maintained in SE. By contrast, and consistent with what has been described previously (*Bloodgood et al., 2013*), we detected a highly significant NPAS4-dependent increase in somatic inhibitory synapses in tissue from mice housed in EE (*Figure 1E and F*; WT: 277.3 ± 14.2 AU, KO: 209.4 ± 6.8 AU, p<0.001, two-way ANOVA with Bonferroni post hoc test). Surprisingly, analyzing superficial and deep sublayers in separate two-way ANOVAs revealed that the NPAS4-dependent increase in inhibitory synapses was significant only within superficial, and not deep, CA1, despite comparable experience-driven NPAS4

expression in both sublayers (*Figure 1G*; values for SE and EE, respectively: superficial – WT: 60.5 ± 4.5 AU, 107.8 ± 14.0 AU, KO: 57.8 ± 15.7 AU, 60.0 ± 7.7 AU, Genotype: $F_{(1,37)}=19.30$, p<0.0001, Housing: $F_{(1,37)}=11.85$, p=0.0014, Interaction: $F_{(1,37)}=17.78$, p=0.0002; deep – WT: 147.5 ± 7.3 AU, 163.4 ± 8.7 AU, KO: 143.5 ± 5.4 AU, 157.2 ± 4.7 AU, Genotype: $F_{(1,37)}=1.89$, p=0.18, Housing: $F_{(1,37)}=5.47$, p=0.025, Interaction: $F_{(1,37)}=0.018$, p=0.89, superficial WT vs KO for EE p<0.001, p>0.05 for all other comparisons, two-way ANOVAs with Bonferroni post hoc tests).

Does this sublayer-specific change in immunofluorescence translate into a sublayer-specific change in synaptic response? To answer this question, the CA1 region of the hippocampus was infected with AAV-Cre-GFP to generate a sparse population of *Npas4* KO neurons within a larger population of WT neurons. Four days after virus injection, animals were housed in SE or EE for 4–7 days and then acute hippocampal slices were prepared. Simultaneous whole-cell voltage clamp recordings were obtained from neighboring WT and *Npas4* KO PNs within superficial or deep sub-layers of CA1. Evoked inhibitory postsynaptic currents (eIPSCs) were elicited by focal stimulation of axons in the pyramidal cell layer and monosynaptic inhibitory currents were isolated by bath applica-tion of N-methyl-D-aspartate (NMDA) and α-amino-3-hydroxy-5-methyl-4-isoxazolepropionic acid (AMPA) receptor antagonists (*Figure 1H,I,K, and L* 10 μM CPP and NBQX). Consistent with our immunohistochemistry data, we measured no systematic difference in eIPSC amplitudes between WT and KO neurons in mice from SE in either superficial or deep PNs (*Figure 1H and K*; Superficial – WT: 1059.9 ± 198.9 pA, KO: 1119.8 ± 192.5 pA, t=1.01, p=0.33, paired t-test, Deep – WT: 952.5 ± 200.7 pA, KO: 959.4 ± 139.55, W=−6.00, p=0.79, Wilcoxon Signed-rank test). However, housing mice in an EE resulted in considerably larger eIPSCs in WT neurons relative to neighboring *Npas4* KO neurons when the pair of PNs was localized to superficial CA1 (*Figure 1I*; WT: 1035.3 ± 196.9 pA, KO: 715.2 ± 116.1 pA, t=2.27, p=0.04, paired t-test). No significant difference in eIPSC amplitude was measured when recording from neighboring WT and KO neurons in deep CA1 from enriched mice (*Figure 1L*; WT: 1146.4 ± 150.6 pA, KO: 1011.1 ± 194.5 pA, t=0.54, p=0.60, paired t-test). Importantly, housing mice in an enriched environment significantly increased the nor-malized difference in eIPSC amplitudes between WT and KO PNs in superficial CA1 (*Figure 1J*; SE: −0.082 ± 0.06, EE: 0.12 ± 0.09, U=54.00, p=0.038, Mann-Whitney U Test), but not in deep CA1 (*Figure 1M*; SE: −0.035 ± 0.11, EE: 0.11 ± 0.11, U=46.00, p=0.18, Mann-Whitney U Test). Thus, experience-driven NPAS4 expression increases somatic inhibitory currents preferentially in superficial CA1 PNs.

## NPAS4 exclusively regulates CCKBC, not PVBC, synapses in CA1

Recent work has revealed new aspects of the precision with which basket cells innervate postsynap-tic targets, including basket cell subtype-specific preferences for PNs in superficial or deep CA1. Inhibition originating from CCKBCs is stronger onto superficial PNs in comparison to those in deep CA1 (*Valero et al., 2015*), while PVBCs more strongly inhibit PNs in deep CA1 (*Lee et al., 2014*) However, it is unclear whether these synaptic preferences are influenced by expression of NPAS4 in postsynaptic PNs.

To determine whether the NPAS4-dependent increase in somatic inhibition observed in superfi-cial CA1 is attributed to CCKBC or PVBC synapses exclusively or to inhibitory synapses in general, we took advantage of key molecular differences between these populations of neurons in order to visualize inhibitory synapses. Synapses that are made by CCK- and PVBCs can be distinguished immunohistochemically via the mutually exclusive expression of the presynaptic cannabinoid recep-tor (CB1R) and the calcium-binding protein parvalbumin (PV), respectively. Thus, we visualized the inhibitory synapses made by CCK-s or PVBCs by staining sections with antibodies recognizing VGAT, gephyrin, and CB1R (*Figure 2A and B*) or PV (*Figure 3A and B*) and quantified the triple overlap to measure synapses made by the respective cell type. Importantly, neither *Cnr1* nor *Pvalb* genes, encoding CB1R or PV, respectively, appear to be direct targets of NPAS4 (*Lin et al., 2008*). Furthermore, only a fraction of CCK- and PV-expressing neurons induce NPAS4 in response to depo-larization (*Spiegel et al., 2014*). Finally, within-animal comparisons of *Npas4* WT and KO hemi-spheres control for potential experience-dependent differences in CB1R- or PV-expression levels (*Donato et al., 2013*).

We compared immunofluorescence corresponding to CCKBC synapses in WT and NPAS4 KO hemispheres of mice from SE and EE in superficial and deep CA1. Immunofluorescence was equiva-lent between the *Npas4* WT and KO hemispheres, in both superficial and deep CA1, when mice

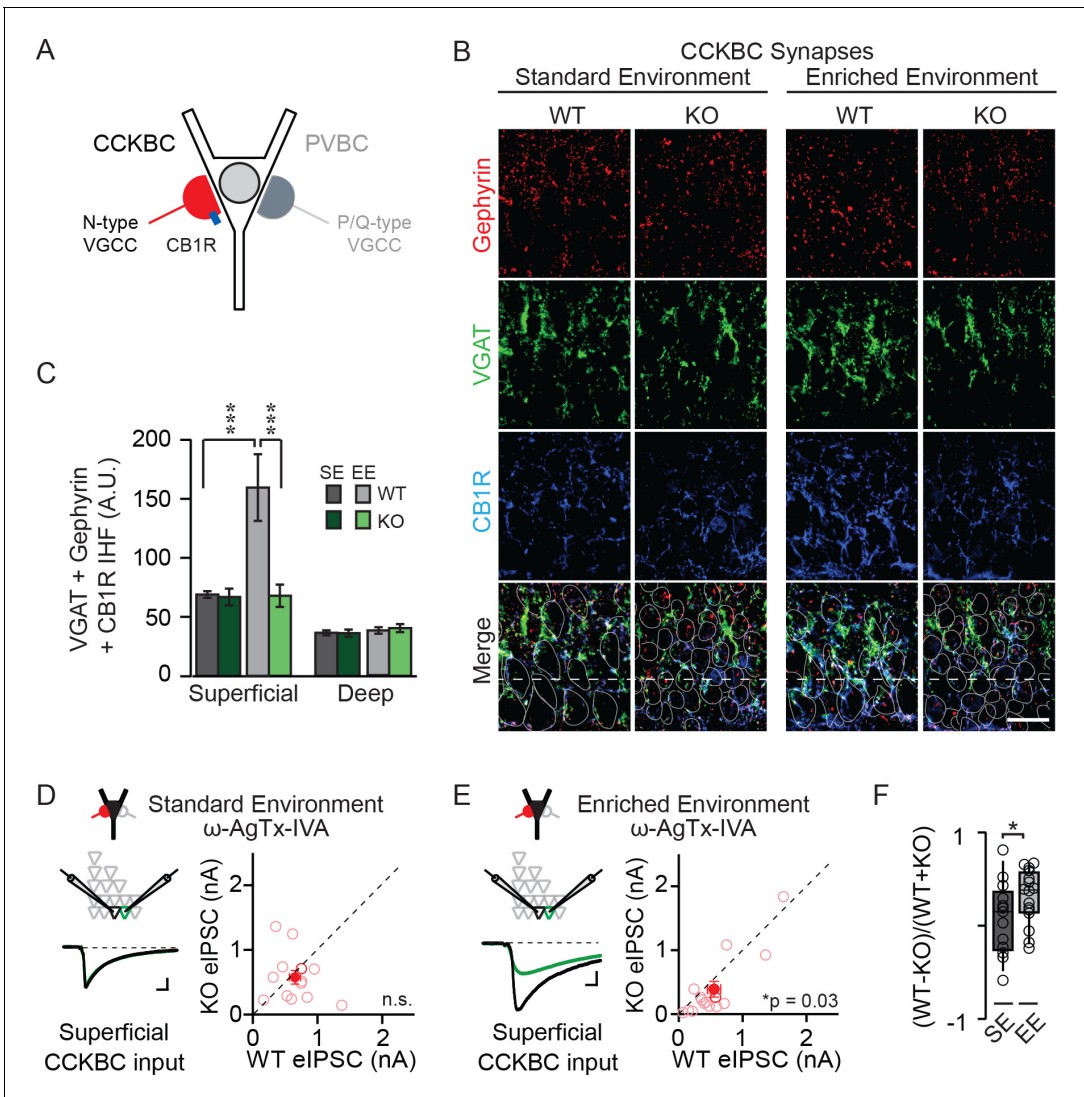

**Figure 2.** NPAS4 regulates CCKBC input onto PNs in superficial CA1. (A) Schematic representation of PN with CCKBC and PVBC synaptic input. CCKBC boutons contain CB1Rs (*blue*) and utilize N-type VGCCs for neurotransmission. (B) Representative images of tissues stained with antibodies against gephyrin (*red*), VGAT (*green*) or CB1R (*blue*) and merged IHC from WT and *Npas4* KO hemispheres from mice housed in SE and EE. Gray dotted lines in merged images represent cell body outlines in the WT condition (mRFP expression) or cell nuclei in the *Npas4* KO condition (Cre-GFP expression). (C) Quantification of CCKBC synapses (overlap of gephyrin, VGAT, and CB1R) in superficial and deep CA1 of WT and *Npas4* KO hemispheres from mice housed in SE and EE (SE: n = 7 sections from a total of 3 mice, EE: n = 4–6 sections from a total of 3 mice). *** indicates p<0.001. (D) ω-AgTx-IVA (0.3 µM) is used to isolate synaptic release from CCKBCs. Example eIPSCs from WT (*black*) and *Npas4* KO (*green*) PNs in superficial CA1 (*left*) of mice housed in SE. Pairwise comparison of eIPSCs recorded in neighboring WT and *Npas4* KO neurons (*right*, n = 13 pairs). (E) As in (D) but from mice housed in EE. Pairwise comparison of eIPSCs recorded in neighboring WT and *Npas4* KO neurons (*right*, n = 16 pairs). (F) Normalized CCKBC eIPSCs recorded from WT and KO pairs across SE and EE for superficial CA1. * indicates p<0.05.

DOI: https://doi.org/10.7554/eLife.35927.006

The following source data and figure supplements are available for figure 2:

**Source data 1.** NPAS4 regulates CCKBC input onto PNs in superficial CA1.

DOI: https://doi.org/10.7554/eLife.35927.009

**Figure supplement 1.** Experience-induced *Npas4* does not affect CCKBC input onto deep CA1 PNs.

DOI: https://doi.org/10.7554/eLife.35927.007

**Figure supplement 1—source data 1.** Experience-induced Npas4 does not affect CCKBC input onto deep CA1 PNs.

DOI: https://doi.org/10.7554/eLife.35927.008

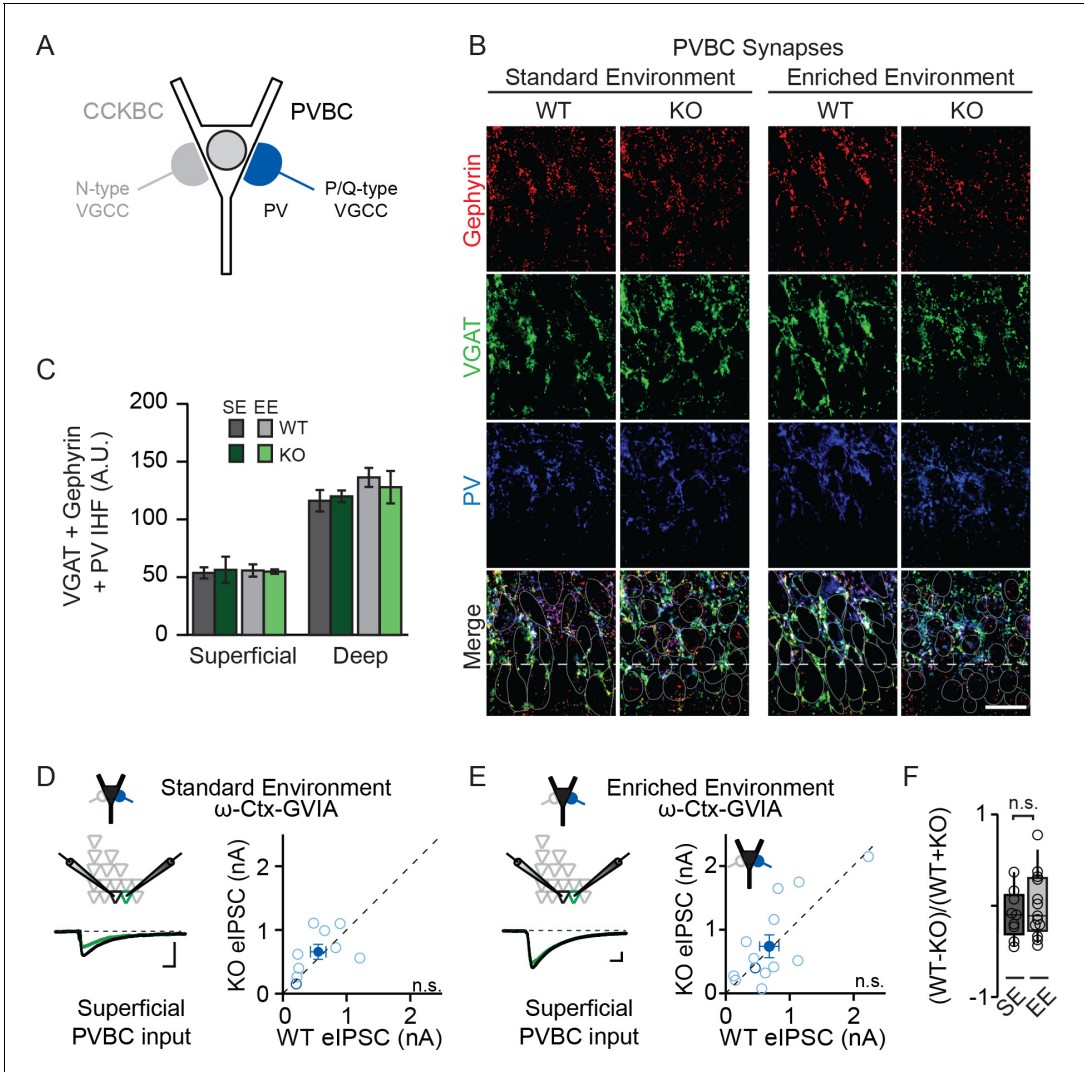

**Figure 3.** NPAS4 does not regulate PVBC input onto PNs in CA1. (A) Schematic representation of PN with CCKBC and PVBC synaptic input. PVBCs express PV (*blue*) and utilize P/Q-type VGCCs for neurotransmission. (B) Representative images of tissues stained with antibodies against gephyrin (*red*), VGAT (*green*) or PV (*blue*) and merged IHC from WT and *Npas4* KO hemispheres from mice housed in SE and EE. Gray dotted lines in merged images represent cell body outlines in the WT condition (mRFP expression) or cell nuclei in the *Npas4* KO condition (Cre-GFP expression). Scale bar = 20 μm. (C) Quantification of PVBC synapses (overlap of gephyrin, VGAT, and PV) in superficial and deep CA1 of WT and *Npas4* KO hemispheres from mice housed in SE and EE (SE: n = 3–4 sections per condition from a total of 3 mice; EE: n = 5 sections per condition from a total of 3 mice). (D) ω-Ctx-GVIA (1 μM) is used to isolate synaptic release from PVBCs. Example eIPSCs from WT (*black*) and *Npas4* KO PNs (*green*) in superficial CA1 (*left*) of mice housed in SE. Pairwise comparison of eIPSCs recorded in neighboring WT and *Npas4* KO neurons (*right*, n = 9 pairs). (E) As in (D) but from mice housed in EE. Pairwise comparison of eIPSCs recorded in neighboring WT and *Npas4* KO neurons (*right*, n = 13 pairs). (F) Normalized PVBC eIPSCs recorded from WT and KO pairs across SE and EE for deep CA1.

DOI: https://doi.org/10.7554/eLife.35927.010

The following source data and figure supplements are available for figure 3:

**Source data 1.** NPAS4 does not regulate PVBC input onto PNs in CA1.
DOI: https://doi.org/10.7554/eLife.35927.013

**Figure supplement 1.** Experience-induced NPAS4 does not affect PVBC input onto deep CA1 PNs.
DOI: https://doi.org/10.7554/eLife.35927.011

**Figure supplement 1—source data 1.** Experience-induced NPAS4 does not affect PVBC input onto deep CA1 PNs.
DOI: https://doi.org/10.7554/eLife.35927.012

were housed in SE (*Figure 2B and C*; SE: superficial – WT: 68.93 ± 3.09 AU, KO: 66.85 ± 7.65 AU; deep – WT: 37.91 ± 3.00 AU, KO: 36.47 ± 2.23 AU, p>0.05 for all comparisons, two-way ANOVA with Bonferroni post hoc tests). Increased sensory experience associated with EE, however, led to a significant increase in immunofluorescence in superficial CA1 from WT hemispheres that was not present in the KO hemispheres (*Figure 2B and C*; EE: superficial – WT: 159.51 ± 28.30 AU, KO: 67.87 ± 9.45 AU, p<0.001, two-way ANOVA with Bonferroni post hoc tests). No significant change was detected in deep CA1 (EE: deep – WT: 40.43 ± 3.38 AU, KO: 36.24 ± 3.05 AU, p>0.05, two-way ANOVA with Bonferroni post hoc tests). Using a two-way ANOVA for each sublayer, we found a significant effect of genotype and housing as well as a significant interaction effect in superficial CA1 (*Figure 2B and C*; Genotype: $F_{(1,20)}$=9.08, p=0.007; Housing: $F_{(1,20)}$=8.67, p=0.008; Interaction: $F_{(1,20)}$=8.29, p=0.009, two-way ANOVA). By contrast, no significant effect of genotype, housing, or interaction was observed for deep CA1 (*Figure 2B and C*; Genotype: $F_{(1,20)}$=0.94, p=0.34; Housing: $F_{(1,20)}$=0.16, p=0.70; Interaction: $F_{(1,20)}$=0.22, p=0.64, two-way ANOVA). These data suggest that activity-driven NPAS4 expression impacts CCKBC synapses and supports the sub-layer specificity observed when evaluating all somatic inhibitory synapses.

In order to determine whether this change in the immunofluorescent detection of CCKBC synapses translates into changes in functional connectivity, we sought to measure inhibition from CCKBCs onto CA1 PNs directly. Stimulation of axons in the cell body layer produces an eIPSC that is a mixture of CCK- and PVBC inputs. These cell types utilize distinct subtypes of voltage-gated calcium channels (VGCCs) to trigger neurotransmitter release (N- and P/Q-type, respectively (*Figure 2A*) (*Hefft and Jonas, 2005*; *Poncer et al., 1997*), enabling the pharmacological isolation of CCKBC neurotransmission by blocking P/Q-type VGCCs (300 nM ω−agatoxin IVA [ω−AgTx-IVA], *Mintz et al., 1992*). Juvenile mice were stereotaxically infected with AAV-Cre-GFP in order to generate a sparse population of *Npas4*-knockout neurons in CA1. After four days, mice were housed in SE or EE for an additional 4–7 days and acute slices were prepared as above. In superficial CA1 of mice from SE, when P/Q-type VGCCs were antagonized and eIPSCs were evoked by electrical stimulation of axons in the pyramidal cell layer, eIPSCs originating from CCKBCs were of equivalent amplitudes in *Npas4* KO PNs and neighboring WT PNs (*Figure 2D*; WT: 778.7 ± 158.8 pA, KO: 960.1 ± 161.2 pA, t=1.01, p=0.33, paired t-test). In superficial CA1 of mice from EE, eIPSCs originating from CCKBCs were ~30% smaller in *Npas4* KO neurons than in neighboring WT neurons (*Figure 2E*; WT: 552.47 ± 104.75 pA, KO: 389. 80 ± 121.90 pA, W=84, p=0.032, Wilcoxon Signed-rank Test). Importantly, housing mice in an EE significantly increased the normalized difference in CCKBC eIPSC amplitudes between WT and KO PNs in superficial CA1 relative to that seen in mice in SE (*Figure 2F*; SE: 0.081 ± 0.10, EE: 0.31 ± 0.07, t=1.18, p=0.041, unpaired t-test). Similar recordings made from PNs in deep CA1 of mice from EE revealed a trend towards WT neurons having larger eIPSC, although the difference was not significant (*Figure 2—figure supplement 1*; WT: 1402.8 ± 341.2 pA, KO: 1081.1 ± 231.9 pA, W=59, p=0.068, Wilcoxon Signed-rank Test). Thus, novel sensory experiences drive an *Npas4*-dependent increase in CCKBC input onto PNs in superficial CA1.

NPAS4-dependent regulation of CCKBC inputs does not preclude the possibility that NPAS4 expression may also regulate PVBC inputs. To test this, we performed experiments that are analogous to those described above, but with immunostaining for PV and pharmacological isolation of PVBC neurotransmission by antagonizing N-type VGCCs (1 μM ω−conotoxin GVIA [ω−Ctx-GVIA], *Figure 3A*) to prevent neurotransmission from CCKBCs. We did not detect any experience- or NPAS4-dependent change in the PVBC synapses that were visualized immunohistochemically (*Figure 3B and C*; SE: superficial – WT: 53.68 ± 4.88 AU, KO: 56.34 ± 13.13 AU; deep – WT: 116.15 ± 9.31 AU, KO: 119.94 ± 5.85 AU; EE: superficial – WT: 55.76 ± 5.30 AU, KO: 54.86 ± 1.94 AU; deep – WT: 136.32 ± 8.25 AU, KO: 127.93 ± 14.02 AU; p>0.05 for all comparisons, two-way ANOVA with Bonferroni post hoc tests). When tested using two-way ANOVA for each sublayer, there was no significant effect of genotype, housing, or interaction effect in either superficial or deep CA1 (*Figure 3B and C*; Superficial CA1; Genotype: $F_{(1,13)}$=0.01, p=0.91; Housing: $F_{(1,13)}$=0.00, p=0.98; Interaction: $F_{(1,13)}$=0.06, p=0.081, two-way ANOVA; Deep CA1; Genotype: $F_{(1,1)}$=0.04, p=0.84; Housing: $F_{(1,13)}$=1.68, p=0.22; Interaction: $F_{(1,13)}$=0.31, p=0.58, two-way ANOVA). Moreover, in mice from both SE and EE, we observed no difference in eIPSC amplitudes recorded from neighboring superficial *Npas4* WT and KO PNs when transmission from PVCBs was pharmacologically isolated (*Figure 3D and E*; SE – WT: 554.5 ± 121.6 pA, KO: 655.7 ± 117.8 pA, t=0.84,

p=0.43, paired t-test; EE– WT: 679.55 ± 150.68 pA, KO: 738.63 ± 179.41 pA, W=−13.00, p=0.65, Wilcoxon Signed-rank test). The normalized difference in PVBC eIPSC amplitudes between WT and KO PNs in mice housed in EE were also equivalent to those for mice housed in SE (*Figure 3F*; SE: −0.1 ± 0.09, EE: 0.002 ± 0.1, t=0.70, p=0.25, unpaired t-test). Lastly, we recorded no difference in eIPSC amplitudes in deep CA1 PN pairs from mice in EE (*Figure 3—figure supplement 1*; WT: 539.7 ± 163.3 pA, KO: 633.9 ± 149.3 pA, W=−16.00, p=0.52, Wilcoxon Signed-rank test). These results indicate that expression of NPAS4 in active CA1 PNs leads to the selective increase of CCKBC inputs without significantly impacting PVBC synapses.

## Environmental enrichment strengthens connectivity between individual CCKBC–PN pairs but does not affect PVBC–PN pairs

We next asked whether the EE-driven selective enhancement of CCKBC input was reflected in the strength of unitary inhibitory post-synaptic currents (uIPSCs) measured between individual basket cell – pyramidal neuron pairs. We recorded from pairs of synaptically connected CCKBCs or PVBCs and superficial PNs in mice living in SE and EE (*Figure 4D–I*). Though genetic strategies exist for the visual identification of PV- and CCK-expressing inhibitory neurons, including Cre- and Flp-dependent intersectional strategies (*Basu et al., 2013*; *He et al., 2016*), these methods are incompatible with our use of Cre to manipulate NPAS4 in *Npas4*[f/f] animals as they would also result in the excision of *Npas4* from the inhibitory neurons. To circumvent this limitation, we identified CCKBCs and PVBCs by a combination of their location, electrophysiological signatures, and morphology. Neurons were targeted that had large somata in CA1 or stratum radiatum (*Bartos and Elgueta, 2012*; *Wisden et al., 2002*) and electrical properties consistent with those of a CCK or PV inhibitory neuron identity (*Figure 4A–C*; CCKBC AP full width at half maximum [FWHM]: 0.95 ms ± 0.04, PVBC AP FWHM: 0.70 ms ± 0.04, t=4.40, p=0.0002, unpaired t-test) (*Glickfeld and Scanziani, 2006*; *Wisden et al., 2002*). For PVBC–PN pairs, we measured no difference in uIPSC amplitudes recorded in mice from SE versus EE (*Figure 4E and F*; SE: 326.2 ± 131.1 pA, EE: 245.3 ± 76.86 pA, U=58.00, p=0.50, Mann-Whitney U Test). However, we recorded significantly larger uIPSC amplitudes between CCKBC–PN pairs from mice in EE relative to SE (*Figure 4H and I*; SE: 155.8 pA ± 27.83, EE: 450.6 pA ± 127.3, U=29.00, p=0.04, Mann-Whitney U Test). Thus, EE drives the selective enhancement of synaptic input from individual CCKBCs onto individual superficial CA1 PNs, without affecting inhibition from PVBCs.

## NPAS4 strengthens CCKBC input by increasing the number of synapses made by individual CCKBCs onto a PN

We next sought to determine whether the mechanism underlying the regulation of CCKBC synapses by NPAS4 involves changes in synapse number, synaptic strength, release probability, or a combination of synaptic changes. We sparsely manipulated *Npas4*, as described above, and housed the animals in EE to reveal NPAS4-dependent changes in CCKBC inhibition. Acute slices were prepared and whole-cell current clamp recordings made from putative CCK inhibitory neurons. Recordings were made in the presence of AM251 (5 μM) to antagonize CB1Rs and to remove the confound of tonic activation of the receptor on uIPSC properties. The non-infected (WT) cells that were targeted had large somata in superficial CA1 or stratum radiatum (*Bartos and Elgueta, 2012*; *Wisden et al., 2002*) and electrical properties consistent with a CCK inhibitory neuron identity (*Figure 4A–C*; *Glickfeld and Scanziani, 2006*; *Wisden et al., 2002*). Inhibitory neurons were filled with biocytin through the patch pipette for *post hoc* morphological analysis (*Figure 4—figure supplement 1D–F*).

After identifying an inhibitory neuron with CCK-like electrical properties, we established a whole-cell recording from a synaptically connected WT or KO PN in superficial CA1 and measured the uIPSC evoked in response to a single AP (*Figure 5A and B*, *Figure 4—figure supplement 1A*). CCK inhibitory neurons are themselves a heterogeneous population of neurons in CA1, comprised of basket cells and dendrite-targeting cells including Schaffer collateral-associated (SCA) and perforant path-associated inhibitory neurons (*Booker et al., 2017*; *Cope et al., 2002*; *Klausberger et al., 2005*; *Pawelzik et al., 2002*; *Vida et al., 1998*). To eliminate the dendrite-targeting cells, which have physiological properties that are largely indistinguishable from those of CCKBCs (*Cope et al., 2002*) from our analysis, we excluded pairs for which the uIPSC success rate was less than 60%

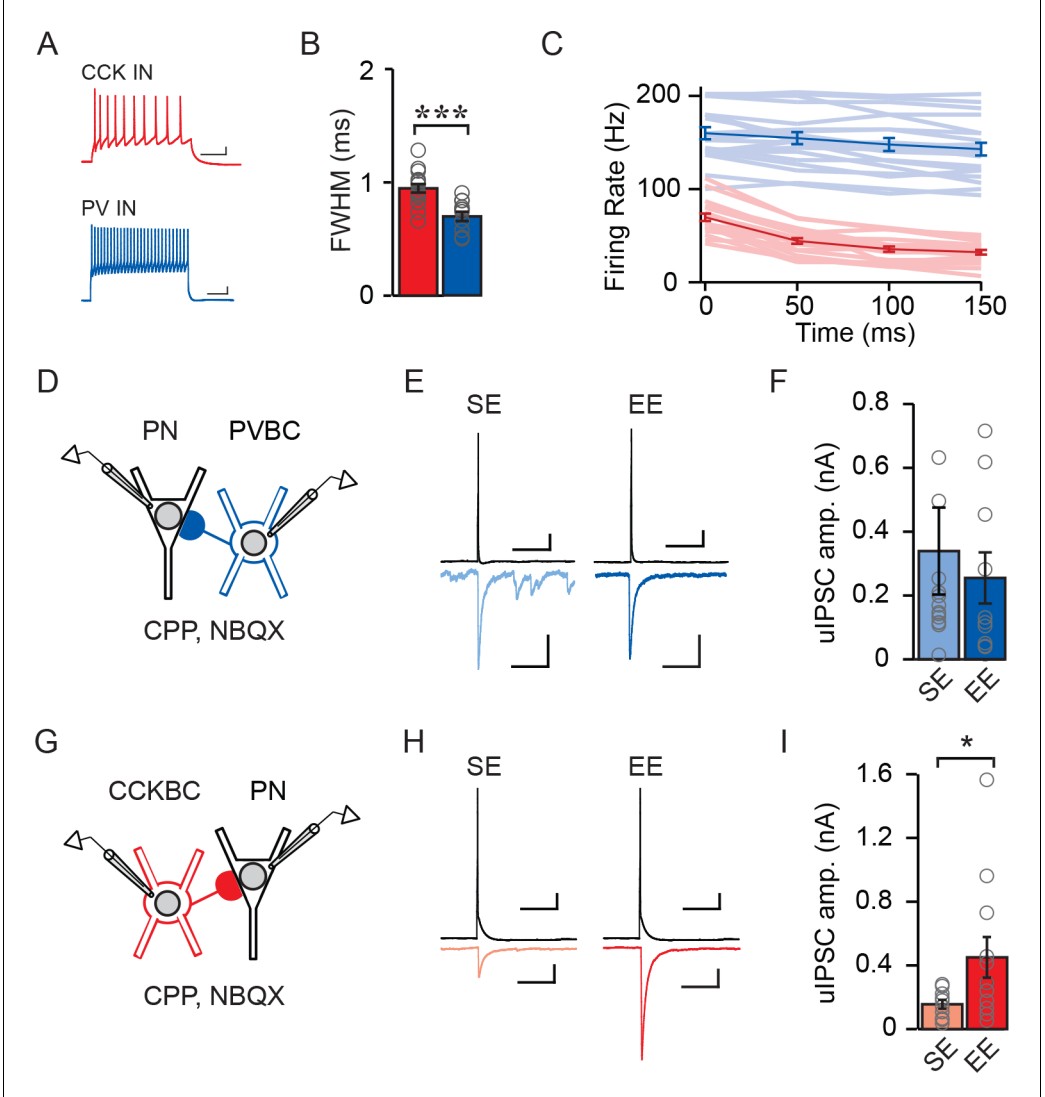

**Figure 4.** Enriched environment increases inhibition of CA1 PNs by CCKBCs but not PVBCs. (A–C) Electrophysiological characteristics of PV inhibitory neurons (INs) and CCK INs: (A) example spike trains (scale bar = 50 ms, 10 mV), (B) AP FWHM n=17 for PV INs, n=20 for CCK INs), *** indicates p<0.001, and (C) AP frequency adaptation over time (n=17 for PV INs, n=19 for CCK INs). (D) Schematic of recording from synaptically connected PVBC and WT PN pairs for panels (E–F). Whole-cell patch clamp recordings were established from PVBCs and synaptically connected WT PNs in mice from SE or EE. (E) Examples of PVBC APs (*top*) and PN uIPSCs (*bottom*) recorded from mice in SE (*light blue*) and EE (*dark blue*). Scale bars: *top* = 50 ms, 10 mV, *bottom* = 50 ms, 100 pA. (F) Average uIPSC amplitudes measured from PVBC-WT PN pairs from mice in SE (*light blue*) and EE (*dark blue*) (SE: n = 14 pairs, EE: n = 10 pairs). Open circles represent individual data points. (G) Schematic of recording from synaptically connected CCKBC and WT PN pairs for panels (H–I). Whole-cell patch clamp recordings were established from CCKBCs and synaptically connected WT PNs in mice from SE or EE. (H) Examples of CCKBC APs (*top*) and PN uIPSCs (*bottom*) recorded from mice in SE (*pink*) and EE (*red*). Scale bars: *top* = 50 ms, 10 mV, *bottom* = 50 ms, 100 pA. (I) Average uIPSC amplitudes recorded from CCKBC–PN pairs from mice in SE (*pink*) and EE (*red*) (SE: n = 10 pairs, EE: n = 12 pairs).

DOI: https://doi.org/10.7554/eLife.35927.014

The following source data and figure supplement are available for figure 4:

**Source data 1.** Distinguishing CCKBCs from CCK SCA interneurons.

DOI: https://doi.org/10.7554/eLife.35927.016

**Figure supplement 1.** CCKBCs were differentiated from dendritic Schaffer collateral-associated (SCA) interneurons by synaptic properties and morphology.

DOI: https://doi.org/10.7554/eLife.35927.015

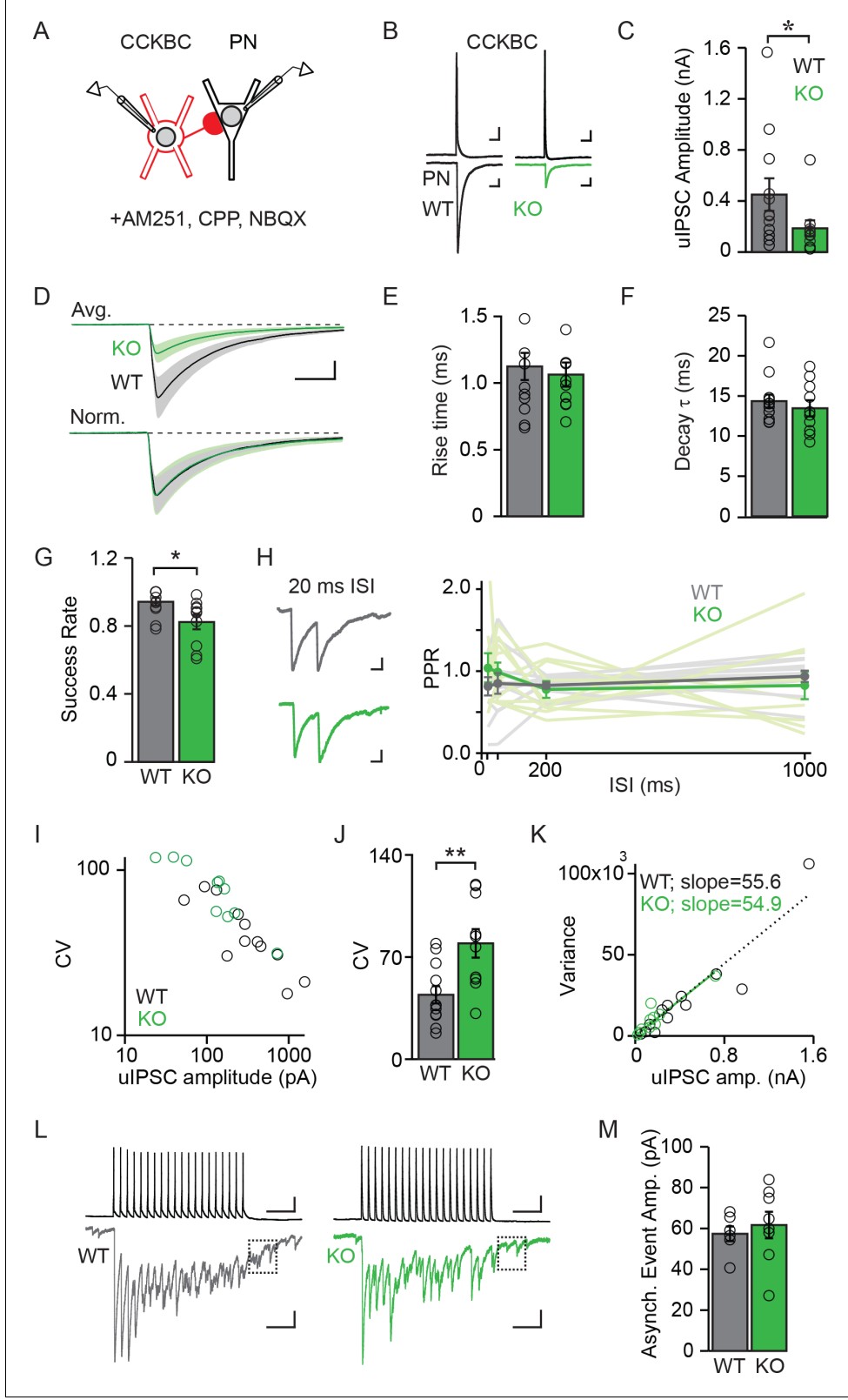

**Figure 5.** NPAS4 regulates the number of CCKBC synapses onto PNs but does not alter synaptic properties. (**A**) Schematic of recording from synaptically connected CCKBC and WT or *Npas4* KO PN pairs. (**B**) Examples of CCKBC APs (*top*) and PN uIPSCs (*bottom*) recorded from WT and *Npas4* KO PNs. Scale bars: *top* = 25 ms, 10 mV; *bottom* = 25 ms, 50 pA. (**C**) Average uIPSC amplitudes recorded from CCKBC–PN pairs for WT and *Npas4* KO

*Figure 5 continued on next page*

*Figure 5 continued*

PNs. (D) Average uIPSC amplitudes recorded from CCKBC–PN pairs for WT and *Npas4* KO PNs (*top*) and normalized by amplitude (*bottom*). Scale bars = 10 ms, 100 pA. (E) 10–90% uIPSC rise times from CCKBC–PN pairs with WT and *Npas4* KO PNs. (F) Decay time constants (τ) from CCKBC–PN pairs with WT and *Npas4* KO PNs. (G) Success rate for CCKBC-PN pairs between WT and *Npas4* KO PNs. Open circles represent individual data points. (H) *Left:* example paired-pulse uIPSCs from WT (*black*) and *Npas4* KO (*green*) PNs recorded from CCKBC–PN pairs with a 20 ms inter-spike interval (ISI). Scale bar = 10 ms, 50 pA. *Right:* paired-pulse ratios (PPRs) for ISIs of 20 ms, 50 ms, 200 ms, and 1000 ms between CCKBC–PN pairs with WT and *Npas4* KO PNs. (I) uIPSC amplitude versus coefficient of variation (CV) for CCKBC–PN pairs with WT and *Npas4* KO PNs. (J) Average CV of the uIPSC recorded in WT and *Npas4* KO PNs. (K) uIPSC amplitude and variance of amplitude for CCKBC–PN pairs with WT (*black*) and *Npas4* KO (*green*) PNs. The dotted black line indicates best linear fit for WT ($R^2$ = 0.89) and the solid green line indicates best linear fit for *Npas4* KO ($R^2$ = 0.84) data. (L) Example traces of 20 APs at 40 Hz and the resulting uIPSC trains for CCKBC–PN pairs with WT (*black*) and *Npas4* KO (*green*) PNs. The dashed box indicates the window analyzed for asynchronous release (100 ms after the end of the AP train). Scale bars = 100 ms, 10 mV (top) and 100 ms, 100 pA (bottom). (M) Asynchronous event amplitude measured during the 100 ms following the end of CCKBC AP firing for CCKBC–PN pairs with WT and *Npas4* KO PNs. (C–K) WT: n=12 pairs, KO: n=10 pairs; (M) WT: n=6; KO: n=8. For all panels, data from WT PNs are shown in black/gray; data from KO PNs are shown in green. Data are shown as mean ± SEM. *p<0.05; **p<0.01.

DOI: https://doi.org/10.7554/eLife.35927.017

The following source data and figure supplements are available for figure 5:

**Source data 1.** Properties of CCKBC synapses made onto WT and *Npas4* KO PNs.
DOI: https://doi.org/10.7554/eLife.35927.020

**Figure supplement 1.** CCKBC–PN pairs with WT and KO PNs have similar uIPSC onset times.
DOI: https://doi.org/10.7554/eLife.35927.018

**Figure supplement 1—source data 1.** uIPSC onset times.
DOI: https://doi.org/10.7554/eLife.35927.019

---

(*Figure 4—figure supplement 1B–C*) (*Younts and Castillo, 2014*). These putative dendrite-targeting neurons tended to evoke smaller amplitude uIPSCs with slower rise times (*Figure 4—figure supplement 1A–C*), consistent with currents that are filtered by extensive stretches of dendrite (*Maccaferri et al., 2000*). Last, when possible, the morphology of the presynaptic neurons was reconstructed (*Figure 4—figure supplement 1D–F*) and pairs omitted if the presynaptic neuron did not have a basket cell-like morphology (for example *Figure 4—figure supplement 1F*). On the basis of these criteria, we analyzed the properties of synaptic connectivity between CCKBCs and WT or *Npas4* KO PNs.

We measured significantly different uIPSC amplitudes in postsynaptic WT and KO PNs in response to single APs evoked in a CCKBC. On average, uIPSC amplitudes in WT PNs were more than twice as large as those measured in KO PNs (*Figure 5C and D*; WT: 450.64 ± 127.29 pA, KO: 183.97 ± 62.70 pA, U=29.00, p=0.044, Mann-Whitney U Test), demonstrating that NPAS4 increases the strength of individual CCKBC–PN connections. Despite different uIPSC amplitudes, the rise times (10–90% of peak) and decay time constants (τ) of the currents were indistinguishable between the two genotypes (*Figure 5D–F*; rise time – WT: 1.13 ± 0.10 ms, KO: 1.06 ± 0.09 ms, t=0.44, p=0.67, unpaired t-test; τ – WT: 14.35 ± 0.83 ms, KO: 13.45 ± 1.01 ms, U=49.00, p=0.49, Mann-Whitney U Test), suggesting that the synapses made onto WT and KO PNs have similar postsynaptic receptor compositions (*Gingrich et al., 1995*; *Lavoie and Twyman, 1996*; *Mody and Pearce, 2004*; *Thomson et al., 2000*).

We next sought to determine whether NPAS4 regulates the numbers of synapses (*N*), the presynaptic probability of release (*P*), or the magnitude of the postsynaptic response (quantal amplitude, *Q*). We first determined the rate of successes and failures of transmission from individual CCKBCs onto WT and KO PNs. Although there were few failures recorded in both genotypes, uIPSCs recorded in *Npas4* KO PNs had a significantly lower success rate than did WT neurons (*Figure 5G*; WT: 0.94 ± 0.02, KO: 0.82 ± 0.04; U=22.00, p=0.01, Mann-Whitney U Test; 60–100 trials per connected pair, APs evoked at 0.1 Hz). This difference in success rate could be explained in one of two ways; NPAS4 could regulate *P* for individual CCKBC synapses, resulting in the measurement of a higher average success rate for CCKBC–WT PN pairs. Alternatively, NPAS4 could regulate the number of synapses made between CCKBC–PN pairs, such that a higher success rate would be recorded

from pairs that have a greater number of synaptic contacts between them, even if all synapses had the same $P$. To test whether NPAS4 might regulate release probability at individual CCKBC synaptic contacts, we measured the postsynaptic responses to pairs of APs and calculated paired pulse ratios (PPRs). Changes in this type of short-term plasticity are generally attributed to shifts in $P$ (*Regehr, 2012*). However, we detected no effect of genotype on PPRs as well as no interaction between genotype and PPR at various inter-spike-intervals (*Figure 5H*; 20 ms – WT: 0.81 ± 0.11, KO: 1.04 ± 0.18, 50 ms – WT: 0.85 ± 0.13, KO: 0.99 ± 0.12, 200 ms – WT: 0.83 ± 0.04, KO: 0.78 ± 0.10, and 1000 ms – WT: 0.94 ± 0.07, KO: 0.83 ± 0.17; Genotype: $F_{(1,57)}=0.77$, p=0.39; Inter-spike-interval: $F_{(3,57)}=0.12$, p=0.94; Interaction: $F_{(3,57)}=0.64$, p=0.59), suggesting that the probability of release at individual synapses made onto WT and KO neurons is unchanged. Similarly, the latency to uIPSC onset after AP firing by the CCKBC was unchanged between the two PN genotypes (*Figure 5—figure supplement 1*; WT: 1.65 ± 0.12 ms, KO: 1.66 ± 0.17 ms, U=51.00, p=0.80, Mann-Whitney U Test), providing further evidence for similar presynaptic organization between synapses converging on *Npas4* WT and KO neurons (*Boudkkazi et al., 2007*, *2011*; *Sabatini and Regehr, 1999*).

If release probability at CCKBC synapses onto WT and KO neurons is indistinguishable, an NPAS4-mediated increase in the number of contact sites made by individual CCKBCs onto PNs is the most parsimonious mechanism to account for the lower success rate of uIPSCs in *Npas4* KO neurons (*Del Castillo and Katz, 1954*). To confirm this and to investigate possible changes in $Q$, we calculated the coefficient of variation (CV) and variance of the uIPSC amplitude recorded from each PN and compared this to the mean uIPSC amplitude. Comparing the CV for each PN's uIPSC amplitude to the mean response revealed a strong negative correlation across all cells, with smaller-amplitude uIPSCs having larger CVs (*Figure 5I*). The average CV for WT PNs was half that measured from KO neurons (*Figure 5J*; WT: 44.17 ± 5.90, KO: 79.40 ± 9.80, t=3.199, p=0.005, unpaired t-test), indicating that the difference in uIPSC amplitudes between WT and *Npas4* KO neurons is due to a change in $N$ and unlikely to reflect differences in $Q$ (*Berninger et al., 1999*; *Kerchner and Nicoll, 2008*; *Le Bé et al., 2007*). Moreover, variance–mean analysis of uIPSC amplitudes shows a linear relationship for both CCKBC–WT PN and CCKBC–KO PN pairs (*Figure 5K*; WT: $R^2=0.89$, KO: $R^2=0.84$) (*Foster and Regehr, 2004*). Strikingly, the linear fits of the WT and KO data sets are essentially superimposable (slope – WT: 55.55 ± 4.73, KO: 54.91 ± 5.46), confirming that the probability of release is unchanged and providing an indirect measurement of the quantal amplitude as ~55 pA in both genotypes (*Reid and Clements, 1999*).

CCKBCs generate significant asynchronous release, which is quantal in nature (*Daw et al., 2009*; *Hefft and Jonas, 2005*), thus we measured the amplitude of asynchronous events allowing us to measure Q directly. A series of 20 APs were evoked at 40 Hz in the CCKBC and the amplitude of asynchronous events quantified during the first 100 ms after the last spike (*Figure 5L*). These amplitudes were comparable between WT and KO neurons (*Figure 5M*; WT: 57.37 ± 3.62 pA, KO: 61.67 ± 6.50 pA, U=18.00, p=0.48, Mann-Whitney U Test) and in close agreement with our estimate of $Q$ from the variance–mean analysis. Thus, we conclude that NPAS4 regulates the number of synapses made by an individual CCKBC onto a PN and does not significantly change the probability of release at individual synapses or the quantal amplitude. On the basis of our measurements of $Q$ and uIPSC amplitudes, individual CCKBCs make, on average, eight synapses onto WT (1–29 synapses) and three onto *Npas4* KOs neurons (1–13 synapses) in animals exposed to enriched environments.

## Sensory experience enhances DSI expression in CA1 PNs through an NPAS4-dependent mechanism

Membrane depolarization triggers the production of endocannabinoids by PNs, which act retrogradely by binding presynaptic CB1Rs, resulting in the transient suppression of inhibitory transmission (DSI) (*Wilson and Nicoll, 2001*). CCKBCs are notable for their expression of CB1Rs and for their cannabinoid-mediated plasticity (*Glickfeld and Scanziani, 2006*). We thus considered the possibility that animals housed in an EE, leading to the formation of more CCKBC synapses, might have more prominent DSI than those maintained in SE. CA1 of *Npas4^{f/f}* mice was sparsely infected with AAV-Cre-GFP and mice were housed in SE or EE as described above. Spontaneous IPSCs were recorded from WT and *Npas4* KO PNs, and DSI was induced by switching into current clamp and triggering 30 APs at 25 Hz (*Figure 6A and B*) (*Dubruc et al., 2013*). We compared the percent suppression of spontaneous IPSC integrated current after DSI (DSI magnitude) in WT and KO PNs from

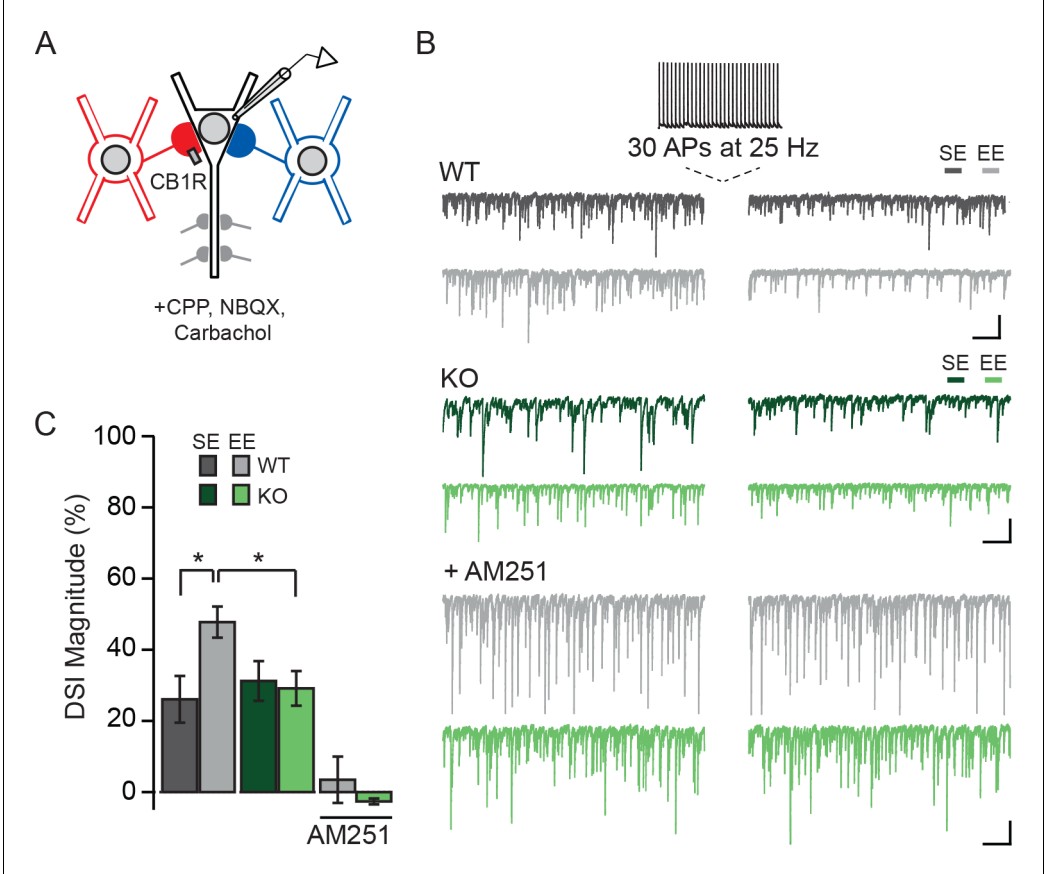

**Figure 6.** Experience-driven NPAS4 expression enhances the magnitude of DSI in PNs. (**A**) Schematic of recording configuration; somatic and dendritic inhibitory inputs are represented. Whole-cell patch clamp recordings were obtained from WT (shown) or *Npas4* KO PNs. Spontaneous IPSCs were recorded before and after inducing DSI by firing 30 APs at 25 Hz in PN. (**B**) Example traces from the experiment described in (**A**) from WT (*gray*) and *Npas4* KO (*green*) PNs recorded in slices from mice in SE, EE, and EE in the presence of the CB1R antagonist AM251 (5 μM). (**C**) DSI magnitude (% reduction in charge after DSI induction) in WT and KO PNs from mice taken from SE, EE, and EE recorded in the presence of AM251 (SE: n=15 WT and 14 KO PNs, EE: n = 14 WT and 13 KO PNs, EE with AM251: n = 11 WT and 11 KO PNs). All scale bars = 500 ms, 200 pA. Data are shown as mean ± SEM. *p<0.05.

DOI: https://doi.org/10.7554/eLife.35927.021

The following source data is available for figure 6:

**Source data 1.** DSI in WT and *Npas4* KO PNs.

DOI: https://doi.org/10.7554/eLife.35927.022

mice in both housing conditions. In WT PNs from mice housed in SE, spontaneous IPSCs were suppressed by approximately 25% after the spike train (DSI magnitude, *Figure 6C*). Notably, housing mice in EE resulted in a nearly two-fold increase in DSI magnitude relative to SE in WT PNs (*Figure 6B and C*, DSI magnitude in WT – SE: 26 ± 7%, EE: 48 ± 4%, p<0.05, two-way ANOVA with Bonferroni post hoc test). As NPAS4 expression in PNs is required for the experience-driven increase in CCKBC synapse number, we hypothesized that experience-dependent enhancement of DSI would also require NPAS4. Indeed, the magnitude of DSI measured in *Npas4* KO PNs from mice living in SE and EE was comparable to that measured in WT PNs from mice in SE (*Figure 6B and C*; DSI magnitude, KO – SE: 31 ± 6%, EE: 29 ± 5%, EE WT vs EE KO p<0.05, two-way ANOVA with Bonferroni post hoc test) and there was a significant interaction effect of genotype and housing (*Figure 6C*; Genotype: $F_{(1,52)}=1.45$, p=0.23, Housing: $F_{(1,52)}=2.90$, p=0.09, Interaction: $F_{(1,52)}=4.27$, p=0.04, two-way ANOVA). Finally, DSI was completely prevented by bath application of the CB1R antagonist AM251 (*Figure 6B and C*; DSI magnitude, WT: 3 ± 7%, KO: 3 ± 1%, WT baseline integrated current

vs integrated current after DSI: W=−26.00, p=0.28, Mann-Whitney U Test; KO baseline integrated current vs integrated current after DSI: W=9.00, p=0.65, Mann-Whitney U Test), confirming that this effect is due to endocannabinoid-mediated DSI. Together, our results support a model of experience-dependent regulation of CCKBC synapses by postsynaptic NPAS4 expression, and demonstrate that NPAS4 enhances endocannabinoid-mediated plasticity in CA1 of the hippocampus.

## Discussion

The expression of IEG-TFs is routinely used to label task-relevant neurons, but we have little insight into how these molecules shape neuronal information processing and contribute to future representations of an animal's environment. *Npas4* is the first known example of an IEG-TF that regulates inhibitory synapses, presenting an opportunity for detailed mechanistic investigation of IEG-TF's role in flexibly adjusting circuit operations. Determining the identity of the inhibitory synapses that are regulated by NPAS4 is critical for understanding how this IEG-TF could alter circuit function. In addition, uncovering the precise way in which inhibition is enhanced provides a basis for future studies aimed at uncovering the molecular mechanisms by which NPAS4 regulates inhibitory synapses.

Here, we have taken advantage of the mutually exclusive receptor and channel expression and electrophysiological signatures of PV- and CCKBCs to reveal the identity of somatic inhibitory synapses that are regulated by NPAS4. We demonstrate with multiple, complementary lines of evidence that experience-driven NPAS4 expression recruits synapses made by CCKBCs, but neither environmental enrichment nor NPAS4 significantly affects synapses made by PVBCs in CA1 (*Figures 2–4*). The conclusions regarding cell-type specificity are strongly supported by immunostaining for CCK- and PVBC synapses, comparisons of pharmacologically isolated eIPSCs from CCK- and PVBCs between WT and *Npas4* KO neurons, and recordings of uIPSCs between CCK- and PVBCs and postsynaptic WT PNs in mice from SE and EE. Each experiment supports the conclusion that experience-dependent NPAS4 expression increases inhibition from CCKBCs, and not PVBCs, but it is possible that changes in PVBC synapses are beyond our detection.

This experience-driven, NPAS4-dependent CCKBC synapse phenotype is most prominent in superficial CA1, where CCKBCs synapse most extensively. Thus NPAS4 likely enhances the gradient of CCKBC inhibition along the superficial to deep axis, and our finding reinforces the idea that superficial and deep CA1 PNs form separate microcircuits (*Danielson et al., 2016*; *Geiller et al., 2017*). Moreover, we find that the NPAS4-regulated gene program increases the number of CCKBC synapses, yet their synaptic properties are indistinguishable from the CCKBC synapses that exist prior to NPAS4 expression (*Figure 5*). Intriguingly, these findings indicate that CCKBCs and PNs communicate through a unique signaling pathway that is dynamically established by the expression of an IEG-TF in the postsynaptic PN. The next challenge will be to elucidate the molecular mechanisms through which experience-driven NPAS4 in PNs specifically communicates with CCKBCs to signal the need for more synapses. Recent work has revealed that the formation of CCKBC synapses relies on postsynaptic expression of the dystroglycan complex (*Früh et al., 2016*), whereas formation of PVBC synapses does not. Dystroglycans, and the proteins that associate with them, are compelling candidates through which NPAS4 may regulate CCKBC synapses, although others may yet be identified.

In comparison to PVBCs, CCKBCs are less well-studied. A distinguishing characteristic of CCKBCs is the sensitivity of their synapses to cannabinoids, and consequently their susceptibility to dynamic regulation by the postsynaptic PN through DSI. Here, we have uncovered an experience-driven enhancement of DSI that requires NPAS4 expression (*Figure 6*), generating several intriguing hypotheses about how the expression of this IEG-TF might shape PN network dynamics. For example, our data imply that PNs in which NPAS4 is expressed will be subject to enhanced CCKBC inhibition, thus reducing CA1 PN spiking during low to moderate network activity. However, when PN activity levels surpass the threshold for triggering cannabinoid production, leading to the temporary suppression of transmission from CCKBCs through DSI, a temporal window will open in which the PN can disinhibit itself, generate more APs, and lower the threshold for plasticity. More broadly, a function of NPAS4 expression may be to increase the signal and reduce the noise within the local microcircuit by refining PN firing (*Bartos and Elgueta, 2012*). As DSI of CCKBC synapses can also facilitate the induction of long-term potentiation (LTP) in PNs (*Carlson et al., 2002*), this window of

disinhibition may facilitate LTP at inputs active during DSI and help to coordinate the plasticity of common inputs to NPAS4-expressing neurons. These hypotheses simply require NPAS4 to regulate DSI by increasing the number of CCKBC synapses, effectively increasing the ratio of somatic inputs onto a PN that are susceptible to DSI, but our experiments cannot rule out the possibility that NPAS4 is also a direct regulator of the DSI pathway.

There is accumulating evidence in support of a role for CCKBCs in shaping hippocampal circuit function. Indeed, abnormal wiring of CCK inhibitory neurons disrupts the spatial coherence of place fields (*Del Pino et al., 2017*), and the opposing gradients of inhibition from PV- and CCKBCs along the superficial to deep axis of the hippocampus underlies the heterogeneity of PN spiking associated with sharp wave ripples (*Valero et al., 2015*). Thus, NPAS4-dependent recruitment of CCKBC inputs may play a role in the spatial coherence of place fields and increase the difference in firing patterns observed during active and resting behavior states.

Notably, exposure of animals to EE induced NPAS4 in similar numbers of PNs localized to superficial or deep CA1, yet we did not detect a significant effect of EE or a requirement of NPAS4 in regulating somatic inhibition in the deep sublayer. This sublayer specificity may indicate that NPAS4 regulates target genes that selectively recruit CCKBC boutons to both superficial and deep PNs, but that a preexisting preference of CCKBC axons for superficial CA1 is the limiting organizational feature. Alternatively, different types of behavioral manipulations may reveal a more significant NPAS4-mediated inhibitory synapse phenotype in deep CA1 PNs. It is also worth noting that while we do not observe a significant change in somatic inhibition of PNs in deep CA1, NPAS4 expression in these cells may regulate other populations of synapses, such as those made by dendrite-targeting INs. The current study describes a role for NPAS4 in regulating somatic inhibition specifically provided by CCKBCs to superficial CA1 PNs, however NPAS4 has also been reported to decrease inhibitory synapses in the proximal apical dendrites of CA1 PNs (*Bloodgood et al., 2013*). A thorough investigation into the nature of NPAS4-dependent regulation of dendritic inhibitory synapses is a crucial next step toward understanding how sensory experience shapes information processing by hippocampal PNs.

# Materials and methods

**Key resources table**

| Reagent type (species) or resource | Designation | Source or reference | Identifiers | Additional information |
|---|---|---|---|---|
| Gene (*Mus musculus*) | *Npas4* | NA | Gene ID: 225872 | |
| Genetic reagent (*Mus musculus*) | *Npas4$^{f/f}$* | Michael Greenberg (HMS) | | |
| Genetic reagent (Adeno-associated virus) | *Cre-GFP* | Penn Vector Core | AAV1.hSyn.HI.eGFP-Cre.WPRE.SV40 | |
| Genetic reagent (Adeno-associated virus) | *mRFP* | Penn Vector Core | AAV1.hSyn.TurboRFP.WPRE.rBG | |
| Antibody | anti-Npas4 | Michael Greenberg (HMS) | | 1 to 1000 |
| Antibody | anti-Gephyrin | Synaptic Systems | RRID:AB_2619834 | 1 to 200 |
| Antibody | anti-VGAT | Synaptic Systems | RRID:AB_887873 | 1 to 350 |
| Antibody | anti-CB1R | Synaptic Systems | RRID: AB_2619970 | 1 to 500 |
| Antibody | anti-parvalbumin | Abcam | RRID:AB_298032 | 1 to 1000 |
| Antibody | anti-NeuN | Synaptic Systems | RRID:AB_2619988 | 1 to 1000 |

Further information and requests for resources and reagents should be directed to lead contact, Brenda Bloodgood (blbloodgood@ucsd.edu).

## Animal husbandry and handling

Animals were handled according to protocols approved by the UC San Diego Institutional Animal Care and Use Committee, which were in accordance with federal guidelines. The animal lines used

were wildtype (WT; C57BL/6J, JAX000664) and *Npas4*[f/f] (*Lin et al., 2008*). Both female and male mice were used. All experiments were performed on animals between postnatal days 21–28 (P21–28) before weaning. For experiments in which mice were injected with virus followed by housing in an enriched environment (*Figures 2–6*), four days after surgery (P17) animals (dam and pups) were moved to a larger cage containing a running wheel, hut, tunnel, and several other objects. To maximize novelty, new objects were introduced and the cage was rearranged every other day. All experiments were conducted on mice housed in an EE for 4–7 days or on mice from SE as indicated.

## Stereotaxic viral injection surgeries

All surgeries were performed according to protocols approved by the UC San Diego Institutional Animal Care and Use Committee and were in accordance with federal guidelines. Stereotaxic viral injection surgeries were performed on P14 mice. Animals were administered Flunixin (2.5 mg/kg) subcutaneously pre-operatively and post-operatively every 12 hr for 72 hr. Animals were deeply anesthetized with isoflurane for the duration of the surgery (initially 3–4% in $O_2$, then maintained at 2%) and body temperature was maintained at 37°C. The fur covering the scalp was shaved and the scalp was cleaned with three iterations of betadine and 70% ethanol before an incision was made to expose the skull. A small burr hole was drilled through the skull over the CA1 region of the hippocampus bilaterally (medial/lateral: 3.1 mm; anterior/posterior: −2.4 mm; dorsal/ventral: 2.8 mm and 2.9 mm below the dura) and virus was injected (350 nL at each dorsal/ventral site for a total of 700 nL; 150 nL min$^{-1}$). Each virus was diluted 2:1 in phosphate-buffered saline (PBS). Three minutes post-injection, the needle was retracted, the scalp sutured and the mouse was recovered at 37°C before being returned to its home cage.

## Virus production

AAV-Cre–GFP was custom produced by the UNC Vector Core with a plasmid provided by M During (Ohio State University). AAV-RFP was a stock virus produced by the Penn Vector Core (AAV1.hWyn. TurboRFP.WPRE.rBG).

## Acute slice preparation

Transverse hippocampal slices were prepared from *Npas4*[f/f] mice (P21–28) 7–15 days after stereotaxic injection of Cre-GFP AAV into CA1. Animals were anesthetized briefly by inhaled isoflurane and decapitated. The cerebral hemispheres were removed and bathed for three minutes in a cold slushy of choline-based dissection solution containing (in mM): 110 choline-Cl, 25 NaHCO$_3$, 1.25 Na$_2$HPO$_4$, 2.5 KCl, 7 MgCl$_2$, 25 glucose, 0.5 CaCl$_2$, 11.6 ascorbic acid, 3.1 pyruvic acid and equilibrated with 95% $O_2$/5% $CO_2$. Blocking cuts were made to isolate the portion of the cerebral hemispheres containing the hippocampus and the tissue was transferred to a slicing chamber containing choline artificial cerebrospinal fluid (choline-ACSF). Slices (300 μM) were cut with a Leica VT1000s vibratome (Leica Instruments) and transferred to a recovery chamber with ACSF consisting of (in mM): 127 NaCl, 25 NaHCO$_3$, 1.25 Na$_2$HPO$_4$, 2.5 KCl, 2 CaCl$_2$, 1 MgCl$_2$, 25 glucose, saturated with 95% $O_2$/5% $CO_2$. Slices were recovered for 30 min at 31° C and maintained at room temperature for the duration of the experiment (4–6 hr).

## Electrophysiology and pharmacology

For experiments performed in tissue from AAV-Cre-GFP-injected mice, infection density varied with distance from the injection site and slices were selected in which ~5–25% of neurons were seen to be infected on the basis of GFP expression as assessed by eye before recordings. Whole-cell voltage-clamp recordings were obtained from CA1 pyramidal neurons and inhibitory neurons visualized with infrared differential interference contrast (IR-DIC) microscopy. Neurons were clamped at −70 mV. During recordings, slices were perfused with ACSF (2–4 mL/ min) bubbled with 95% $O_2$/5% $CO_2$ and heated to 31°C. For pyramidal neuron recordings, patch pipettes (open pipette resistance 2–4 MΩ) were filled with an internal solution containing (in mM) 147 CsCl, 5 Na$_2$-phosphocreatine, 10 HEPES, 2 MgATP, 0.3 Na$_2$GTP and 2 EGTA (pH = 7.3, osmolarity = 300 mOsm) and supplemented with QX-314 (5 mM), except in DSI experiments. For experiments in which eIPSCs in WT and *Npas4* KO pyramidal cell pairs were recorded, extracellular stimulation of local axons within specific lamina of the hippocampus was delivered by current injection through a theta glass stimulating electrode

that was placed in the center of the relevant layer (along the radial axis of CA1) and within 100–300 µm laterally of the patched pair. eIPSCs were pharmacologically isolated with CPP (10 µM) and NBQX (10 µM) in all experiments as well as with ω-AgTx-IVA (0.3 µM) or ω-Ctx-GVIA (1 µM) where indicated. For inhibitory neuron- pyramidal cell connected pair experiments, inhibitory neurons were patched with pipettes filled with an internal solution containing (in mM): 147 K-gluconate, 20 KCl, 10 $Na_2$-phosphocreatine, 10 HEPES, 2 Na-ATP, 0.3 Na-GTP, 5 $MgCl_2$, 0.2 EGTA, and 3% biocytin (Sigma Aldrich B4261) (pH = 7.3, osmolarity = 300 mOsm). Interneurons were held in the current clamp at a resting membrane potential of −70 mV. For DSI experiments, spontaneous inhibitory activity was induced in acute slices with carbachol (5 µM). DSI was induced in pyramidal cells by triggering 30 action potentials at 25 Hz.

## Biocytin visualization and reconstructions

Interneurons recorded with an internal solution containing 3% biocytin were labeled using a diamino-benzene (DAB) reaction, as previously described with modifications (*Marx et al., 2012*). Briefly, cells were held for 15–30 min. After gently detaching from the cell, slices were placed in 4% PFA and fixed overnight. After fixation, slices were stored in PBS until processing. All the following steps were carried out at 4°C on a rotating platform and all washes were 10 min, unless otherwise noted. Slices were washed 6x in 100 mM phosphate buffer (PB; consisting of $NaH_2PO_4^-$ and $NaPO_4^-$, pH 7.4), incubated for 20 min in PB + 3% $H_2O_2$, washed 4x in PB, then incubated overnight in a permeabilization buffer (3% Triton X-100, 2% normal goat serum [NGS] in PB). The next day, slices were washed 1x in PB, incubated for 2 hr in a 'pre-incubation' buffer (0.5% Triton X-100, 0.5% NGS in PB), then incubated in a biotinylation buffer (pre-incubation buffer + ABC solutions [ThermoScientific 32050]; 1% of 'Reagent A' Avidin + 1% 'Reagent B' biotinylated horseradish peroxidase) for 2 hr. Slices were then washed 3x in PB, 2x in Tris Buffer (TB; 50 mM Tris base, pH 7.4), incubated for 10 min at room temperature in DAB solution 1 (1% Imidazole, 1 tablet/2 mL DAB [Sigma Aldrich D5905] in TB), and then incubated in DAB solution 2 (1% imidazole, 1% ammonium nickel sulfate hydrate ($NH_4$) $2Ni(SO_4)_2$, 1 tablet/2 mL DAB, 3% $H_2O_2$, in TB) for 2–10 min at room temperature, or until the slices turned visibly dark purple. Slices were immediately washed in PB for 1 min, followed by 2x washes in PB. Slices were then mounted on slides (Superfrost/Plus, Fisher Scientific) and air dried overnight. The following day, slices were dehydrated and cleared with the following steps (6 min each): 30% ethanol, 50% ethanol, 70% ethanol, 96% ethanol, twice in 100% ethanol, and three times in xylene. Slices were then cover slipped with Krystalon (EMD Millipore) and dried overnight in the chemical fume hood. Biocytin-filled inhibitory neurons were reconstructed on an Olympus DSU microscope using Neurolucida software (MBF Bioscience).

## Immunohistochemistry

For labeling of PV- and CB1R-positive inhibitory synapses, P14 mice were stereotaxically injected with AAV-Cre-GFP into CA1 of the right hemisphere and AAV-RFP into CA1 of the left hemisphere. After 4 days of recovery from surgery and 4–7 days in an enriched environment or standard housing, mice were anesthetized briefly with isoflurane and decapitated. Hippocampi were rapidly dissected in ice-cold dissection media consisting of (in mM): 1 $CaCl_2$, 5 $MgCl_2$, 10 glucose, 4 KCl, 26 $NaHCO_3$, 218 sucrose, 1.3 $NaH_2PO_4 \cdot H_2O$, 30 HEPES. Hippocampi were immediately drop fixed in 4% paraformaldehyde in PBS at 4°C for 2 hr followed by overnight incubation in 30% sucrose in PBS. Cryoprotected tissue was stored in Tissue-Tec O.C.T. at −20°C, sectioned at 20 µM (Leica CM1950 cryostat) and mounted on slides (Superfrost/Plus, Fisher Scientific).

For NPAS4 and inhibitory synapse immunostaining, hippocampal sections were blocked in 5% goat serum and 0.2% Triton X-100 in PBS overnight at 4°C. Sections were incubated in primary antibody overnight at 4°C in blocking solution, washed three times in PBS, and incubated overnight in a species-matched secondary at 4°C, and washed again three times in PBS. Slices were briefly dipped in $ddH_2O$ and cover slipped with Fluoromount (Electron Microscopy Sciences). See 'Key Resources Table' for antibodies and concentrations used for all IHC experiments.

## Confocal imaging

All slices and tissue sections were imaged using an Olympus Fluoview 1000 confocal microscope (× 10/.4, × 20/0.75, and × 60/1.42 [oil] plan-apochromat objectives; UC San Diego School of Medicine

Microscopy Core, supported by NINDS grant NS047101). Identical acquisition parameters were used for all slices or tissues within a single experiment. The levels, contrast, and brightness of confocal images were moderately adjusted in Photoshop CS6 software (Adobe Systems, Inc.) for illustrative purposes using scientifically accepted procedures.

## Image quantification

Confocal images for a particular experiment were subjectively thresholded using ImageJ software and the threshold was kept consistent across images from all conditions obtained for a single experiment. For immunohistochemistry experiments (*Figure 1*), the integrated density (the product of the area and mean grey value, termed the immunohistofluorescence (IHF) of the overlap of the three fluorescent signals, was quantified within regions of interest (ROIs) for superficial and deep CA1 using ImageJ software (National Institute of Health). Puncta were defined as a thresholded fluorescence cluster with an area $\geq 0.05$ µm$^2$. Superficial and deep CA1 ROIs were 25 µM bins (along superficial to deep axis) aligned to the superficial and deep edges of the CA1 stratum pyramidale, respectively (*Lee et al., 2014*). IHF was normalized to cell number within each ROI as determined by DAPI counterstaining.

## Electrophysiology analysis

Electrophysiology data were acquired using ScanImage software (*Pologruto et al., 2003*) and a Multiclamp 700B amplifier. Data were sampled at 10 kHz and filtered at 6 kHz. Off-line data analysis was performed using scripts written in Igor Pro by ALH and BLB (Wavemetrics).

Experiments were discarded if the holding current for pyramidal cells with CsCl-based internal solution was greater than $-500$ pA or if the series resistance was greater than 25 MΩ. In experiments in which direct comparisons were made between two neurons, recordings were discarded if the series resistance differed by more than 20% between the two recordings. All recordings were performed at 31°C.

The amplitudes of eIPSCs and uIPSCs were calculated by averaging the amplitude 0.5 ms before to 2 ms after the peak of the current. Data are shown as positive values for clarity. For connected inhibitory neuron–pyramidal cell paired recordings, paired pulse ratios (PPR) were calculated by recording a template uIPSC for each cell, normalizing it to the peak of the first pulse of the PPR wave, subtracting the template wave from the PPR wave, and then measuring the corrected amplitude of the second peak. Asynchronous event amplitudes were measured for events greater than 15 pA within the 100 ms following the end of a train of 20 APs delivered at 40 Hz to the CCKBC. Asynchronous event amplitudes were only counted for cells in which the spontaneous event frequency recorded for the cell was less than 40% of that of the event frequency recorded during the asynchronous release analysis window. Slopes of the rise times of the uIPSCs were measured by normalizing the uIPSC, then measuring the slope between 10–90% of the uIPSC peak. For DSI experiments, the DSI magnitude was reported as the ratio of the total inhibitory current per second, averaged over 4 s after DSI, to the total inhibitory current per second, averaged for 10 s before DSI induction.

## Statistics

All values are expressed as mean ± SEM. Box plots are displayed as the median (center line), 75–25% (upper and lower box), and 90–10% (whiskers). For all electrophysiology experiments, 'n' refers to the number of cells or pairs recorded per condition and are biological replicates. For immunohistochemistry experiments, 'n' refers to tissue sections used per condition, obtained from a specified number of separate mice and are biological replicates. The aforementioned values can be found in the figure legends. Our sample sizes were not pre-determined and are similar to those reported in the literature. Data collection was not performed blind to the conditions of the experiment and we did not use any specific randomization procedure other than to assign litters of mice to one of two experimental housing conditions in an alternating manner. Analysis was done blind to condition, and when possible, experiments were designed to allow for within-animal comparisons between WT and *Npas4* KO pyramidal cells. All statistical analysis was performed in Prism (GraphPad Software, Inc., La Jolla, CA). Parametric tests were used for data sets with normal distributions by both the D'Agostino-Pearson and Shapiro-Wilk tests. Wilcoxon signed-rank tests or paired t-tests were performed for paired data and Mann-Whitney U Tests or unpaired t-tests were used for unpaired data. In

experiments involving two independent variables, where n was too small to determine distribution, a normal distribution was assumed and two-way analysis of variance (ANOVA) with Bonferroni post-hoc tests were performed. F statistics and degrees of freedom are reported as F(df model, df residual). Statistical significance was assumed when $p < 0.05$. In all figures, *$p < 0.05$, **$p < 0.01$ and ***$p < 0.001$ as determined in Prism software. All figures were generated using Illustrator CS6 software (Adobe Systems, Inc.).

## Acknowledgements

This work was supported by grants from the Pew Charitable Trust (00028631), the Searle Scholars Program (14-SSP-184), the Whitehall Foundation (2013-12-88) (to BLB), and the National Science Foundation (2013154395) (to ALH). We thank G Higerd, A Kazi, K Mendoza, and G Moore for technical assistance. We thank J Isaacson, M Scanziani, T Komiyama and members of the lab for critical reading of the manuscript and thoughtful feedback. We are particularly grateful to J Isaacson for his ongoing and thoughtful discussions during the preparation of this manuscript.

## Additional information

### Funding

| Funder | Grant reference number | Author |
| --- | --- | --- |
| Pew Charitable Trusts | 00028631 | Brenda L Bloodgood |
| Kinship Foundation | 14-SSP-184 | Brenda L Bloodgood |
| Whitehall Foundation | 2013-12-88 | Brenda L Bloodgood |
| National Science Foundation | 2013154395 | Andrea L Hartzell |

The funders had no role in study design, data collection and interpretation, or the decision to submit the work for publication.

### Author contributions

Andrea L Hartzell, Conceptualization, Data curation, Software, Formal analysis, Supervision, Validation, Investigation, Visualization, Methodology, Writing—original draft, Writing—review and editing, Performed all electrophysiology except CCKBC-PN pairs in SE, Analyzed all electrophysiology data, Performed surgeries, immunohistrochemistry and image analysis; Kelly M Martyniuk, Investigation, Writing—review and editing, Performed surgeries and immunohistochemistry; G Stefano Brigidi, Formal analysis, Investigation, Writing—review and editing, Performed immunohistochemistry and image analysis; Daniel A Heinz, Investigation, Performed recordings between CCKBCs and PNs from mice in SE; Nathalie A Djaja, Formal analysis, Investigation, Performed immunohistochemistry and image analysis; Anja Payne, Investigation, Performed surgeries; Brenda L Bloodgood, Conceptualization, Data curation, Formal analysis, Supervision, Funding acquisition, Validation, Visualization, Methodology, Writing—original draft, Project administration, Writing—review and editing

### Author ORCIDs

Andrea L Hartzell  http://orcid.org/0000-0002-6202-6148
Brenda L Bloodgood  http://orcid.org/0000-0002-4797-9119

### Ethics

Animal experimentation: This study was performed in strict accordance with the recommendations and guidance provided by the Research Compliance and Integrity Program and the Institutional Animal Care and Use Committee (IACUC) at UC San Diego. All of the animals were handled according to the approved IACUC protocol, S12254, of UC San Diego. All surgeries and euthanasia were performed under deep isoflurane anesthesia, and every effort was made to minimize suffering.

Decision letter and Author response
Decision letter https://doi.org/10.7554/eLife.35927.025
Author response https://doi.org/10.7554/eLife.35927.026

## Additional files

**Supplementary files**
• Transparent reporting form
DOI: https://doi.org/10.7554/eLife.35927.023

**Data availability**

All data generated or analyzed during this study are included in the manuscript and supporting files.

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
