## [Decision Letter]

Thank you for submitting your article "The IEG *Npas4* recruits CCK basket cell synapses and enhances cannabinoid-mediated plasticity in the mouse hippocampus" for consideration by *eLife*. Your article has been reviewed by three peer reviewers, one of whom is a member of our Board of Reviewing Editors, and the evaluation has been overseen by Gary Westbrook as the Senior Editor. The following individual involved in review of your submission has agreed to reveal their identity: Imre Vida (Reviewer #2). The reviewers have discussed the reviews with one another and the Reviewing Editor has drafted this decision to help you prepare a revised submission.

Summary:

This paper shows that enriched environment exposure causes increases in CCK basket cell synapses onto pyramidal cells in CA1, and associated increases in depolarization-induced suppression of inhibition, and that these effects are induced by increased levels of the inducible transcription factor NPAS4. All three reviewers agreed that these are important and significant results and that the data supporting the main results are convincing. However, there are some points that should be clarified to increase the likelihood that these results would have a broad impact. Also, some conclusions should be toned down (i.e., all conclusions must be based strongly on convincing data, see essential points 2-6 below). The specific substantial concerns are listed below.

Essential revisions:

1) Throughout the manuscript, the authors compare different experimental effects, but it is unclear whether they compared the different experimental effects in the same statistical analysis, as is proper (see Nieuwenhuis et al., Nature Neuroscience 2011). As one example (although this issue persists throughout the paper), in Figure 2C, where the authors refer to an ANOVA, what exactly did they test in the ANOVA? Did they observe a significant interaction between environmental condition and genotype? Related to this point, the authors should report statistics (e.g., F-statistics) and degrees of freedom, not just p-values.

2) Regarding the Abstract and Introduction, last paragraph, Discussion, second paragraph (Selective enhancement of CCKBC inhibition). In the paired recording experiments, the authors do not report if *Npas4* KO affected PVBC-PN unitary IPSCs, after EE or SE exposure. Were experiments of this type carried out at the same time as the reported experiments and not reported (since paired recordings were conducted blindly to interneuron types, and identified with posthoc anatomy)? Without such experiments the conclusions of cell-type selective actions are not strongly supported by the physiological evidence. The sole physiological evidence of selectivity is that the presumed PVBC eIPSC (in conotoxin) is unaffected by *Npas4* KO after EE exposure. However, this is indirect evidence of lack of effect on presumed PVBC synapses. For this reason, without these data, the conclusions regarding cell specificity of these effects should be toned down.

3) The authors' conclusion that EE selectively enhances inhibition mediated by CCKBCs is not supported by direct evidence. This was tested for PVBC-PN synapses (Figure 4—figure supplement 2) showing no change in unitary connectivity of PVBC to PC. But the authors did not test in WT mice if eIPSCs in superficial PNs (ex. Figure 1), pharmacologically isolated IPSC (ex. Figure 2) or unitary IPSCs of CCKBC-PN pairs (ex. Figure 4) were larger after EE relative to SE. The results with KO showed that a component of these IPSCs is *Npas4*-dependent after EE, but it is not clear if this "new" *Npas4*-dependent component contributes to an enhancement of synaptic inhibition, or not. For this reason, the strength of these conclusions should be toned down.

4) Experience-driven NPAS4 expression is similar in superficial and deep PNs, yet anatomical evidence indicates changes in inhibitory synaptic connectivity (VGAT, gephyrin, CB1R) only in superficial PNs. The physiological evidence about NPAS4 effects in deep and superficial PNs is not so clear. The authors do not systematically report (and presumably did not test) *Npas4* KO effects in SE and EE, on IPSCs in superficial and deep PNs (i.e., subsection “NPAS4 underlies an experience-dependent enhancement of somatic inhibition in superficial CA1”, last paragraph and Figure 1: the effects of KO in deep PNs after SE exposure was not tested?; subsection “NPAS4 exclusively regulates CCKBC, not PVBC, synapses in CA1”, fourth paragraph and Figure 2: the effects of KO in sup and deep PNs in SE condition was not tested?; subsection “NPAS4 exclusively regulates CCKBC, not PVBC, synapses in CA1”, last paragraph and Figure 3: the effects of KO in sup PNs after SE, or in deep PNs after SE and EE, were not tested?; subsection “NPAS4 strengthens CCKBC input by increasing the number of synapses made by individual CCKBCs onto a PN”, first paragraph and Figure 4: was the effect of KO on unitary CCKBC IPSCs only observed after EE? Is the effect of KO absent in mice exposed to SE?). In the absence of these data, the conclusions, again, should be toned down.

5) Subsection “NPAS4 strengthens CCKBC input by increasing the number of synapses made by individual CCKBCs onto a PN”, fourth paragraph, and Figure 5: The rate of success of transmission in CCKBC-PN pair recordings was reduced after *Npas4* KO. This seems as the closest direct measure of presynaptic probability of release at those synapses that the authors have. Thus it would seem that their conclusion of unaltered properties of unitary IPSCs (subsection “NPAS4 strengthens CCKBC input by increasing the number of synapses made by 232 individual CCKBCs onto a PN”, fifth and last paragraphs, Discussion, second paragraph and elsewhere) is incorrect, as is their conclusion of no change in presynaptic release probability. Although it is not clear how *Npas4* KO postsynaptically would affect presynaptic release, it would seem that the authors cannot rule out such changes.

6) Abstract, Introduction, last paragraph, subsection “Sensory Experience Enhances DSI Expression in CA1 PNs through an *Npas4*-Dependent Mechanism”, Figure 6: With regard to NPAS4 as regulator of plasticity, it is not clear if NPAS4 is a direct or indirect regulator of plasticity because of enhanced spontaneous inhibition. Could the increased magnitude in DSI after EE be due to the increased level of synaptic inhibition after EE? In Figure 6, the level of sIPSC activity seems higher after EE relative to SE in wt. Is this actually the case? Are sIPSC frequency and amplitude different in SE vs EE? Is there a correlation between sIPSC frequency pre-DSI and magnitude of DSI (in SE and EE)? Does normalizing/reducing sIPSC activity after EE to similar levels as SE, restore DSI magnitude? This would seem important to resolve since if so *Npas4* regulation of DSI would be indirect and due to its effect on level of inhibition. These possibilities should be addressed or discussed.

7) Regarding the title, IEG is not necessarily a standard acronym that is understood by the broad scientific audience of *eLife. eLife* tries to avoid all acronyms in the title. Also, "plasticity" may be misleading, and the authors should consider using "modulation" instead.

---

## [Author Response]

Essential revisions:1) Throughout the manuscript, the authors compare different experimental effects, but it is unclear whether they compared the different experimental effects in the same statistical analysis, as is proper (see Nieuwenhuis et al., Nature Neuroscience 2011). As one example (although this issue persists throughout the paper), in Figure 2C, where the authors refer to an ANOVA, what exactly did they test in the ANOVA? Did they observe a significant interaction between environmental condition and genotype? Related to this point, the authors should report statistics (e.g., F-statistics) and degrees of freedom, not just p-values.

We have clarified the statistics throughout the manuscript.

- We tested all data sets for normality and clarified where we ran T tests versus nonparametric equivalents. Our methods for normality testing were added to the Materials and methods section.

- We indicate where we have used a T-test, Mann-Whitney U-test, paired t-test, Wilcoxon signed-rank test, and two-way ANOVA with Bonferroni post hoc tests).

- The conditions being compared, degrees of freedom, and U- and F-statistics are reported as appropriate.

2) Regarding the Abstract and Introduction, last paragraph, Discussion, second paragraph (Selective enhancement of CCKBC inhibition). In the paired recording experiments, the authors do not report if Npas4 KO affected PVBC-PN unitary IPSCs, after EE or SE exposure. Were experiments of this type carried out at the same time as the reported experiments and not reported (since paired recordings were conducted blindly to interneuron types, and identified with posthoc anatomy)? Without such experiments the conclusions of cell-type selective actions are not strongly supported by the physiological evidence. The sole physiological evidence of selectivity is that the presumed PVBC eIPSC (in conotoxin) is unaffected by Npas4 KO after EE exposure. However, this is indirect evidence of lack of effect on presumed PVBC synapses. For this reason, without these data, the conclusions regarding cell specificity of these effects should be toned down.

We have included two additional experiments (Figure 3 and Figure 3—figure supplement 1) exploring the impact of EE and NPAS4 on PVBC synapses. We do not see any evidence of EE or NPAS4-dependent regulation of PVBC synapses made onto PNs in any context. Nonetheless, we do not have sufficient data to completely assert this (i.e. synaptically connected pairs of PVBC and NPAS4 knockout PNs from mice housed in SE and EE) so we have addressed this in the Discussion.

To briefly summarize our findings with respect to PVBCs:

- Figure 3: We show by co-immunostaining for VGAT, gephyrin, and PV that WT and KO hemispheres are indistinguishable in both superficial and deep CA1 from mice housed in SE or EE. Thus, the immunostaining proxies for PVBC synapses do not change with EE or genotype.

- Figure 3:In the presence of conotoxin to isolate PVBC inputs, neighboring WT and KO PNs have similar amplitude evoked IPSCs in superficial CA1 from mice housed in SE and EE. Thus, PVBC mediated inhibition within the superficial sub-layer does not change, even though this is the region where we see both EE and NPAS4 dependent changes in inhibition

- Figure 3—figure supplement 1: In the presence of conotoxin, WT and KO PNs have similar amplitude evoked IPSCs in deep CA1 from mice in housed in EE.

- Figure 4:The amplitude of unitary IPSCs recorded from synaptically connected PVBC–WT PNs pairs from mice housed in SE and EE are indistinguishable, confirming EE does not appreciably change PVBC inputs onto individual PNs.

Collectively these data indicate that PV inputs onto PNs do not appreciably change between SE and EE and loss of NPAS4 also does not have a detectable impact on PVBC synapses isolated pharmacologically. However, we don’t have data looking specifically at synaptically connected PVBC-NPAS4 KO PNs pairs. When we began searching for synaptically connected BC–NPAS4 KO PN pairs, we had already established that EE was not significantly impacting PVBC-PN inhibitory synapses but was changing CCKBC-PN connectivity. Because of this, we focused on CCKBC–PN connections and did not look for synaptically connect PNs when the presynaptic neuron had the electrophysiological profile of a PVBC. As these experiments are quite low yield, it was not possible for us to collect this data set within the 2-month revision period, but this caveat is addressed in the text.

3) The authors' conclusion that EE selectively enhances inhibition mediated by CCKBCs is not supported by direct evidence. This was tested for PVBC-PN synapses (Figure 4—figure supplement 2) showing no change in unitary connectivity of PVBC to PC. But the authors did not test in WT mice if eIPSCs in superficial PNs (ex. Figure 1), pharmacologically isolated IPSC (ex. Figure 2) or unitary IPSCs of CCKBC-PN pairs (ex. Figure 4) were larger after EE relative to SE. The results with KO showed that a component of these IPSCs is Npas4-dependent after EE, but it is not clear if this "new" Npas4-dependent component contributes to an enhancement of synaptic inhibition, or not. For this reason, the strength of these conclusions should be toned down.

We have included new data supporting our assertion that EE selectively enhances inhibition from CCKBCs onto PNs.

- Most directly we now show in Figure 4 that the amplitude of unitary IPSCs recorded from synaptically connected CCKBCs and WT PN pairs more than doubles in animals from EE as compared to SE.

- Figure 2: We now show that in the presence of agatoxin to isolate CCKBC inputs, neighboring WT and KO PNs have similar amplitude evoked IPSCs in superficial CA1 from mice housed in SE. Previously we had shown that in mice housed in EE, the evoked IPSC is larger in the WT PN in comparison to the neighboring KO PN.

- Figure 2—figure supplement 1: We show that in the presence of agatoxin, WT and KO PNs have similar amplitude evoked IPSCs deep CA1 from mice housed in EE, again indicating NPAS4 specifically regulates somatic inhibition onto superficial CA1 PNs.

Thus, EE doubles the unitary IPSC made by CCKBCs onto WT PNs. This phenotype is restricted to superficial CA1 and it not present in NPAS4 KO neurons.

4) Experience-driven NPAS4 expression is similar in superficial and deep PNs, yet anatomical evidence indicates changes in inhibitory synaptic connectivity (VGAT, gephyrin, CB1R) only in superficial PNs. The physiological evidence about NPAS4 effects in deep and superficial PNs is not so clear. The authors do not systematically report (and presumably did not test) Npas4 KO effects in SE and EE, on IPSCs in superficial and deep PNs (i.e., subsection “NPAS4 underlies an experience-dependent enhancement of somatic inhibition in superficial CA1”, last paragraph and Figure 1: the effects of KO in deep PNs after SE exposure was not tested?; subsection “NPAS4 exclusively regulates CCKBC, not PVBC, synapses in CA1”, fourth paragraph and Figure 2: the effects of KO in sup and deep PNs in SE condition was not tested?; subsection “NPAS4 exclusively regulates CCKBC, not PVBC, synapses in CA1”, last paragraph and Figure 3: the effects of KO in sup PNs after SE, or in deep PNs after SE and EE, were not tested?; subsection “NPAS4 strengthens CCKBC input by increasing the number of synapses made by individual CCKBCs onto a PN”, first paragraph and Figure 4: was the effect of KO on unitary CCKBC IPSCs only observed after EE? Is the effect of KO absent in mice exposed to SE?). In the absence of these data, the conclusions, again, should be toned down.

We now include data comparing WT and KO PNs in superficial and deep CA1 from animals in SE and EE (Figure 1). Additionally (and described in the response to points 2 and 3), we detect an EE and NPAS4 dependent change in CCKBC inputs onto superficial CA1 PNs but not deep (Figure 2, Figure 2—figure supplement 1); PVBC are unchanged in every context (Figure 3, Figure 3—figure supplement 1). We also have evaluated uIPSCs from CCKBCs onto WT PNs in SE and see they are smaller than in EE and comparable to uIPSCs recorded from KO PNs in EE (Figure 4). Together these data show that EE and NPAS4 are only impacting somatic inhibition in superficial CA1 PNs and this phenotype emerges from experience and NPAS4-dependent increases in CCKBC inputs.

We do not exclude the possibility that NPAS4 expression in deep CA1 PNs regulates some other aspect of PN function (for example dendritic inhibitory synapses) and this is explicitly included in the Discussion section.

5) Subsection “NPAS4 strengthens CCKBC input by increasing the number of synapses made by individual CCKBCs onto a PN”, fourth paragraph, and Figure 5: The rate of success of transmission in CCKBC-PN pair recordings was reduced after Npas4 KO. This seems as the closest direct measure of presynaptic probability of release at those synapses that the authors have. Thus it would seem that their conclusion of unaltered properties of unitary IPSCs (subsection “NPAS4 strengthens CCKBC input by increasing the number of synapses made by 232 individual CCKBCs onto a PN”, fifth and last paragraphs, Discussion, second paragraph and elsewhere) is incorrect, as is their conclusion of no change in presynaptic release probability. Although it is not clear how Npas4 KO postsynaptically would affect presynaptic release, it would seem that the authors cannot rule out such changes.

The success rate of transmission at a single synapse is directly related to the probability of release. Since each CCKBC axon makes multiple synapses, and transmission from each synapse is independent, the success rate recorded in the postsynaptic neuron conflates P and N. For example, if there are 2 boutons each with p = 0.5, the success rate of detecting any transmission between the cells would be 75%. But if there are 15 synapses each with P = 0.5, the predicted success rate of transmission between the cells would be essentially 1. Our measurements of paired pulse facilitation, a reliable indicator of P, at CCKBC synapses made onto WT and KO PNs are not significantly different between genotypes. This suggests that the number of synapses made onto KO neurons is likely smaller resulting in a lower overall success rate for transmission. We have attempted to communicate this more clearly in the text.

6) Abstract, Introduction, last paragraph, subsection “Sensory Experience Enhances DSI Expression in CA1 PNs through an Npas4-Dependent Mechanism”, Figure 6: With regard to NPAS4 as regulator of plasticity, it is not clear if NPAS4 is a direct or indirect regulator of plasticity because of enhanced spontaneous inhibition. Could the increased magnitude in DSI after EE be due to the increased level of synaptic inhibition after EE? In Figure 6, the level of sIPSC activity seems higher after EE relative to SE in wt. Is this actually the case? Are sIPSC frequency and amplitude different in SE vs EE? Is there a correlation between sIPSC frequency pre-DSI and magnitude of DSI (in SE and EE)? Does normalizing/reducing sIPSC activity after EE to similar levels as SE, restore DSI magnitude? This would seem important to resolve since if so Npas4 regulation of DSI would be indirect and due to its effect on level of inhibition. These possibilities should be addressed or discussed.

We have quantified spontaneous IPSC frequency and amplitude in WT and KO PNs from SE and EE. By 2-way ANOVA we see a significant effect of environment on sIPSC frequency at baseline, with mice in SE having *higher* baseline activity than mice in EE. We observed no effect of genotype on sIPSC frequency at baseline as well as no interaction effect. For sIPSC amplitude at baseline, we observe no effect of housing environment, genotype, or interaction. While higher sIPSC frequency in SE at baseline is surprising, this is hard to interpret in the context of using carbachol to induce spontaneous IPSCs. As a cholinergic agonist, carbachol will increase spontaneous release from multiple cell types. This is especially confounding given that experience-induced *Npas4* has been shown to decrease dendritic inhibitory synapses in mice housed in EE. It is also possible that some component of the cholinergic system in the hippocampus is down regulated by activity induced by environmental enrichment. Because of these considerations, we cannot interpret this piece of information in good faith so we would prefer to not include it in the final manuscript. We do find it to be an extremely interesting observation that we hope to explore further in the future.

To the reviewers’ next point, there is no significant correlation between sIPSC frequency and the magnitude of DSI by linear regression for each data set: SE WT, SE KO, EE WT, and EE KO. We have addressed whether *Npas4* might be regulating DSI indirectly through increasing CCKBC synapses versus directly regulating the DSI pathway in the Discussion.

7) Regarding the title, IEG is not necessarily a standard acronym that is understood by the broad scientific audience of eLife. eLife tries to avoid all acronyms in the title. Also, "plasticity" may be misleading, and the authors should consider using "modulation" instead.

These are points well taken. We have modified the title of the manuscript to, “*Npas4* recruits CCK basket cell synapses and enhances cannabinoid-sensitive inhibition in the mouse hippocampus”.